# Organization of reward and movement signals in the basal ganglia and cerebellum

**Noga Larry** [1,2] ✉, **Gil Zur**[1,2] **& Mati Joshua** [1] ✉

The basal ganglia and the cerebellum are major subcortical structures in the motor system. The basal ganglia have been cast as the reward center of the motor system, whereas the cerebellum is thought to be involved in adjusting sensorimotor parameters. Recent findings of reward signals in the cerebellum have challenged this dichotomous view. To compare the basal ganglia and the cerebellum directly, we recorded from oculomotor regions in both structures from the same monkeys. We partitioned the trial-by-trial variability of the neurons into reward and eye-movement signals to compare the coding across structures. Reward expectation and movement signals were the most pronounced in the output structure of the basal ganglia, intermediate in the cerebellum, and the smallest in the input structure of the basal ganglia. These findings suggest that reward and movement information is sharpened through the basal ganglia, resulting in a higher signal-to-noise ratio than in the cerebellum.

One of the core roles of the brain is to generate movements that increase the chance of gaining rewards. To achieve this, sensory information along with predictions about the reward outcomes of actions is processed into movement. This process is dispersed across multiple brain structures. To understand how reward is used to drive movement, the contribution of each area to the transformation of reward information into a motor command needs to be disentangled. One possible approach is to constrain the computations of each area by identifying area-specific signals. However, in a complex interconnected system such as the brain, specificity is rare, and signals are often found ubiquitously[1–3]. Thus, a system approach that compares signals across populations is better positioned to yield insights into the transformations of signals as they propagate across neural populations.

The basal ganglia and cerebellum are two major subcortical structures that control movement. The basal ganglia are considered to be involved in choices between actions and reward-based learning[4,5]. By contrast, the cerebellum has been hypothesized to be involved in adjusting movement and error-based motor learning[4,6,7]. However, recent findings of reward signals in the cerebellum have challenged this dichotomous division of labor[1,8–12], and imply that the fundamental difference between these structures may not be so straightforward[13,14].

Although reward signals in the basal ganglia and cerebellum exhibit some similarity, they currently cannot be directly compared, since experiments have been conducted on different behaviors, tasks, and species, and in parts of these structures that are involved in the production of different movements.

To overcome the limitations of previous research, we compared signals in the basal ganglia and the cerebellum directly by recording from oculomotor areas in both structures in the same tasks and monkeys. Focusing on oculomotor regions allowed us to build on well-established knowledge on the coding of eye movements[15–17], the connectivity and anatomy of the oculomotor system[18], and the causal relationship between structures and movements[19–23] to study how reward and movement signals are multiplexed in these structures. In the basal ganglia, we recorded from the input structure, the body of the caudate, and the output structure, the substantia nigra pars reticulata (SNpr). The activity of the caudate and the SNpr neurons is modulated by both eye movements and reward, thus supporting the claim that the basal ganglia implement reward-based learning[17,24–26]. In the cerebellum, we recorded from the oculomotor vermis, which is causally involved in the production of saccade and pursuit eye movements[20,21]. The vermis has also been suggested to participate in cognitive and affective processing[27–29], making it an intriguing region

[1]Edmond and Lily Safra Center for Brain Sciences, the Hebrew University, Jerusalem, Israel. [2]These authors contributed equally: Noga Larry, Gil Zur.
✉e-mail: noga.larry@mail.huji.ac.il; mati.joshua@mail.huji.ac.il

of the cerebellum to study the interactions between reward and motor signals. We designed tasks that manipulated eye movement direction and the probability of receiving a reward, which enabled us to examine reward expectations, motor signals, and how they interact.

We found that reward expectation and eye movement direction signals were the most pronounced in the output of the basal ganglia, intermediate in the vermis, and the smallest in the input of the basal ganglia. This suggests that information is sharpened through the basal ganglia to a greater extent than through the cerebellar cortex. These results extend recent findings of reward coding in the cerebellum and provide the first direct comparison with the basal ganglia.

## Results

### Eye movement tasks that manipulate reward probability and movement direction

To compare the coding of eye movements and reward directly in the vermis and the basal ganglia, we recorded from both areas in the same monkeys (Fig. S1, Table S1). The monkeys performed smooth pursuit and saccade tasks in which we manipulated the probability of receiving a reward and the direction of eye movements. The trials began with the appearance of a white spot in the center of the screen (Fig. 1a). The monkeys were required to fixate on the spot, which then changed into one of two colors indicating the probability of the reward that would be delivered upon successful completion of the trial. One color corresponded to a probability of 0.75 (blue in Fig. 1a) and the other to a probability of 0.25 (red). In the pursuit task, after a variable delay, the colored target began to move 20°/s in one of eight directions[30] and the monkey was required to smoothly track the target (Fig. 1b top, Fig. S2). In the saccade task, the target disappeared and reappeared at one of eight 10° eccentric locations. Shortly ($221 \pm 44$ and $194 \pm 37$ ms for Monkeys A and G) after the target jumped, the monkey was required to saccade towards the target (Fig. 1b bottom, Fig. S2). At the end of a successful trial, the target disappeared, and the monkey received a reward with the probability specified by the target color.

We chose reward probabilities of 0.25 and 0.75 since they generated substantial reward expectation differences that were easily noticeable by the monkeys. We confirmed this through a separate task, in which the monkeys were required to select one of two targets based on reward probability associations (see Methods). In this two-target task, the monkeys constantly tracked the $P = 0.75$ target over the $P = 0.25$ target (Fig. 1c, Signed-rank test for the difference from chance: $p_{Monkey\ A}$, $p_{Monkey\ G} < 0.001$). Although there were differences in behavior between conditions in both the saccade and pursuit tasks, these differences were very small (see Methods, Fig. S2), thus allowing us to disambiguate the coding of motor variables from the coding of reward.

We segmented both tasks into three separate epochs. The first epoch was the cue where information about the probability of future reward was first made available. The second was the motion epoch, in which the target moved, and pursuit or saccade movements occurred. The third epoch was the outcome where the reward was either delivered or omitted.

### Direct comparison of neurons in the basal ganglia and vermis

To examine how reward and movements are represented across structures, we recorded from areas in the basal ganglia and cerebellum involved in the generation of pursuit and saccades[15–17,21,31,32]. In the basal ganglia, we recorded from the body of the caudate, an input structure, and the SNpr, an output structure. In the cerebellum, we recorded from the oculomotor vermis. We identified Purkinje cells, the sole output of the cerebellar cortex, based on the presence of complex spikes followed by a pause in the activity of simple spikes[33,34]. Other neurons were classified as putative local neurons that do not project outside the cerebellar cortex.

In each population, we found neurons that responded to the presentation of the reward probability cue (e.g., Fig. 1d–g, Fig. S3, Table 1). To compare these responses to the cue between populations, we calculated the reward probability effect size. Specifically, we used the partial $\omega^2$ ($\omega_p^2$, see Methods). $\omega_p^2$ is an unbiased estimator of the variability explained by an experimental variable, normalized by the same variability plus a noise term[35] (Fig. 1h). The noise term is an estimator of the variability that is unaccounted for by any variable in the experiment; i.e., the trial-by-trial variability within task conditions. $\omega_p^2$ values close to 1 indicate large differences between the levels of a variable relative to the noise within conditions, whereas values close to 0 indicate smaller or no differences between conditions.

Figure 1j shows the reward probability effect size in 50 ms time bins for each example neuron. All neurons showed an increase in effect size after the appearance of the cue when the reward probability information first became available. However, this increase was substantially larger in the SNpr neuron (Fig. 1f). This increase corresponded to the time within the trial when the difference between the probability condition in the SNpr neuron was large and the within-condition variability was small. These examples demonstrate that the effect size reflects the signal-to-noise ratio in activity. Intuitively, a large effect size is correlated with improved decoding of the experimental condition from the neural activity (Fig. S4a–c).

This approach has several advantages. First, $\omega_p^2$ is agnostic to the direction of response. Some neurons in our sample had higher rates when the reward probability was high (for example, Fig. 1e, f) while others had higher rates when the reward probability was low (Fig. 1d, g). All the populations had neurons with higher rates for either the high or the low probability conditions (Fig. S3)[36,37]. Second, the variability explained by additional variables is removed from the noise term. This is critical when comparing populations that multiplex the coding of experimental variables to different extents. Finally, since $\omega_p^2$ is an unbiased estimator, when a variable does not affect the firing rate, $\omega_p^2$ will be 0 on average, in contrast to other effect size measures in which the bias depends on the number of conditions (Fig. 1i).

We also calculated the effect size over the entire trial (see Methods). This yielded a single assessment of the reward probability coding throughout the entire epoch. These whole epoch effect sizes are specified in the caption of Fig. 1d–g. We verified that the effect sizes were robust to different bin sizes (Fig. S4d–i) and that they were not determined by the firing rates of the neurons (Fig. S5).

### The coding of reward probability is more pronounced in the SNpr than in the vermis and the caudate

To estimate the coding of reward probability in the populations around the time of cue presentation, we calculated the reward probability effect size of individual neurons and the average across each population (Fig. 2a). We pooled the data across both tasks since the structure of the task in this epoch was identical and the results were the same for the two tasks. As expected from an unbiased estimator, before the reward probability information was made available, the reward effect size of all populations was close to zero (Fig. 2a, before time 0). After cue presentation, there was a transient increase in the reward probability effect size in all populations. This effect was the most pronounced in the SNpr (Fig. 2b, Permutation Welch's ANOVA test: $p < 0.001$, Permutation Welch's $t$-test: $p_{SNpr-caudate}$, $p_{SNpr-purkinje}$, $p_{SNpr-local} < 0.001$). The increase in reward probability effect size was transient and decreased after the initial increase, although it remained slightly larger than zero in all four populations. It is unlikely that this result was due to the coding of conditioned licking in the cue epoch because these were similar across the different reward probability conditions (Fig. S6).

This analysis confirms recent findings reporting reward-related activity in the cerebellum[1,9–11] and the numerous studies indicating

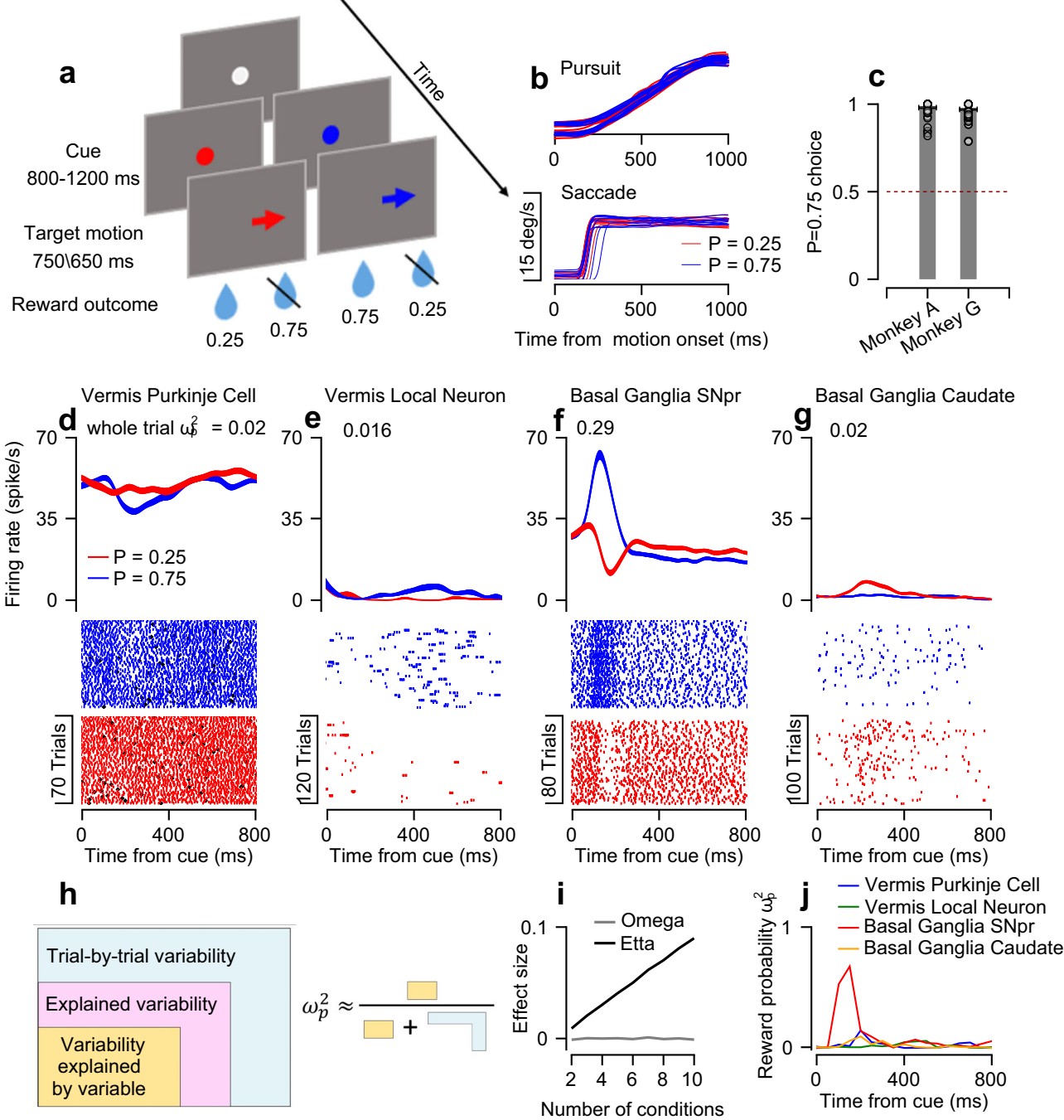

**Fig. 1 | Effect size analysis to quantify the coding of reward probability on a trial-by-trial basis. a** Schematics of the pursuit and saccade eye movement tasks. The spots represent the target and the arrows show the direction of continuous motion (pursuit) or instantaneous jump of the target (saccade). The numbers represent the probability of receiving a reward. **b** Ten example traces of eye position from a single pursuit (top) and saccade (bottom) session aligned to target motion onset. **c** The fraction of trials in which the monkey selected the $P = 0.75$ target on an additional choice task averaged across sessions. Gray dots represent individual sessions (Monkey A: $n = 71$, Monkey G: $n = 87$). The dashed line represents the chance level. All error bars represent the standard error of the mean. **d**–**g** PSTHs (top) and rasters (bottom) aligned to the onset of the cue for the $P = 0.25$ (red) and $P = 0.75$ (blue) conditions in a vermis Purkinje cell (**d**), vermis local neuron (**e**), SNpr neuron (**f**), and caudate neuron (**g**). The black diamonds in (**d**) represent complex spikes recorded in the same Purkinje cell. Error bars on PSTHs represent SEMs over trials. **h** An illustration of the calculation of the $\omega_p^2$ effect size. The parts represent the partition of the trial-by-trial variability into the variance explained by a specific variable (yellow), the variance explained by other variables (pink), and the variance unexplained by any of the task variables (blue). **i** Average $\omega_p^2$ (gray) and partial $\eta^2$ (black) effect sizes as a function of the number of conditions in a simulation in which the neural responses were independent of the condition. **j** Reward probability $\omega_p^2$ effect sizes of neurons in (**d**–**g**) calculated in bins of 50 ms. Entire trial effect sizes that represent the total coding of reward probability throughout the cue epoch are shown in (**d**–**g**).

reward-related activity in the basal ganglia[37–39]. However, beyond these studies, these findings demonstrate that the SNpr responses were substantially stronger than the other populations. The results indicate a strong sharpening of reward signals within the basal ganglia, from the caudate to the SNpr. The finding that the caudate exhibited a smaller average effect size than both vermis populations is surprising given the hypothesis that the basal ganglia are the primary site where reward signals coincide with the coding of movements[4].

**Table 1 | Number of significant neurons**

|  | Reward probability in the cue epoch | Direction in the motion epoch | Both |
|---|---|---|---|
| Caudate | 76 (0.45) | 101 (0.43) | 45 (0.19) |
| SNpr | 106 (0.44) | 199 (0.86) | 95 (0.41) |
| Local cerebellar neurons | 60 (0.44) | 69 (0.51) | 41 (0.3) |
| Purkinje cell simple spikes | 35 (0.38) | 48 (0.53) | 26 (0.28) |

The number and fraction (in parenthesis) of neurons that significantly differentiated between the reward probability conditions during the cue epoch, the target motion direction during the motion epoch, and their intersect.

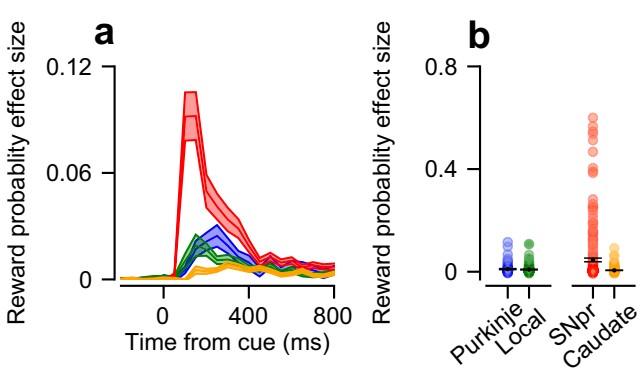

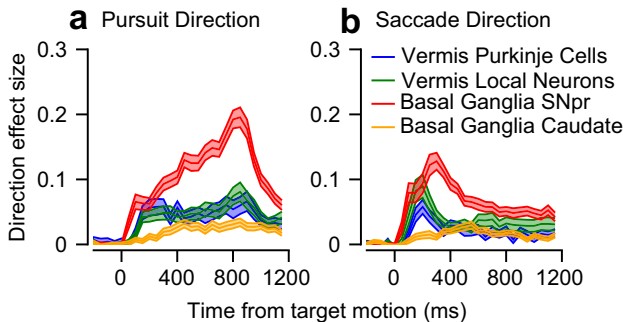

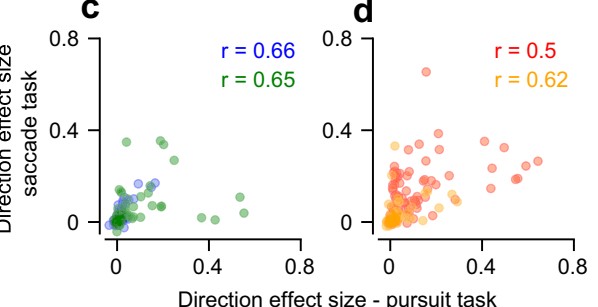

**Fig. 3 | Coding of eye movement direction during pursuit and saccades.**
**a**, **b** Direction effect size during the target motion epoch on the pursuit task (**a**) and the saccade task (**b**). **c**, **d** Each dot represents a single neuron's direction effect size on the pursuit (horizontal) and saccade (vertical) tasks. The *r* values represent the Spearman correlation. **c** shows the vermis populations and **d** the basal ganglia.

**Fig. 2 | Coding of reward probability during the cue epoch. a** Average reward probability effect size in the cue epoch as a function of time. In all the following figures, error bars represent the standard error of the mean across neurons. **b** Each dot represents a single neuron's reward probability effect size in the entire cue epoch.

### The coding of eye movement direction is more pronounced in the SNpr than in the vermis and the caudate

To study how eye movement signals distribute across subcortical networks, we next compared the coding of eye movement directions between populations on the pursuit and saccade tasks (Fig. 3 and Figs. S7, S8). As we focused our analysis on oculomotor regions, many neurons in our sample differentiated between the direction conditions (Table 1). Some neurons had complex temporal dynamics[40] (for example, Fig. S7c). The $\omega_p^2$ effect size analysis allowed us to compare the coding of directions between tasks and populations without averaging in time to calculate the tuning curve or having to define the preferred direction of neurons[41]. We, therefore, calculated the direction effect size of each population and task. All the populations exhibited an increase in direction effect sizes as the target began to move, on both the pursuit and the saccade tasks (Fig. 3a, b). This increase was the most pronounced in the SNpr (Pursuit – Permutation Welch's ANOVA test: $p < 0.001$, Permutation Welch's $t$-test: $p_{SNpr-caudate}$, $p_{SNpr-Purkinje}$, $p_{SNpr-local} < 0.001$; Saccade - Permutation Welch's ANOVA test: $p < 0.001$, Permutation Welch's $t$-test: $p_{SNpr-caudate}$, $p_{SNpr-Purkinje} < 0.001$, $p_{SNpr-local} = 0.15$). On the saccade task, the difference between the SNpr and local vermis population did not reach significance, but the average effect size was larger in the SNpr. The larger average effect size in the SNpr in comparison to the cerebellar output was surprising, since the vermis is thought to be functionally closer to behavior[20,23].

### Common control for saccades and pursuit

A substantial fraction of neurons were recorded on both the pursuit and saccades tasks (see Methods). We used $\omega_p^2$ to revisit the question of common control for saccade and pursuit[42] in the vermis[20] and the

SNpr[32] and examine this question for the first time in the caudate. We found significant correlations between the direction effect sizes on the pursuit and saccade tasks in all four populations (Fig. 3d, e; Spearman correlation: $r_{SNpr} = 0.5$, $r_{caudate} = 0.5$, $r_{Purkinje} = 0.66$, $r_{local} = 0.65$, $p < 0.001$ in all populations). These correlations indicate that in the vermis and both the input and output structures of the basal ganglia, neurons that are tuned to direction during pursuit also tend to be tuned to direction during saccades. Thus, our results support claims of common control for saccades and pursuit in the subcortical networks.

### Coding of reward probability during movement

Our analysis method allowed us to quantify the extent to which reward was coded during movement. One of the main benefits of $\omega_p^2$ is that it takes the variability in the response of neurons that is related to additional experimental variables into account. The noise term in $\omega_p^2$ is defined such that variability that can be explained by experimental variables is not treated as noise (Fig. 1h). This allows us to compare the coding of reward in populations that encode the direction of eye movements to varying extents. When the targets began to move, the reward probability effect size was small in all populations (Fig. 4a). SNpr neurons showed a relatively late increase in the coding of reward probability, which was not observed in any of the other populations. The coding of direction during the movement epoch was substantially larger than the coding of reward probability in all populations. This was indicated by larger direction effect sizes than reward probability effect sizes (Fig. 4b, most dots fall above the identity line, Bootstrap $t$-test: $p < 0.001$ for all populations, see examples in Fig. S7i–l). These results are consistent with previous studies on the basal ganglia[43], cerebellum[11], and frontal eye field[44], implying a general organization of the motor systems in which reward information is mostly represented when it is first available and then only weakly during movement.

While the coding of the reward condition itself was slight, reward expectation may modulate the coding of direction, having an

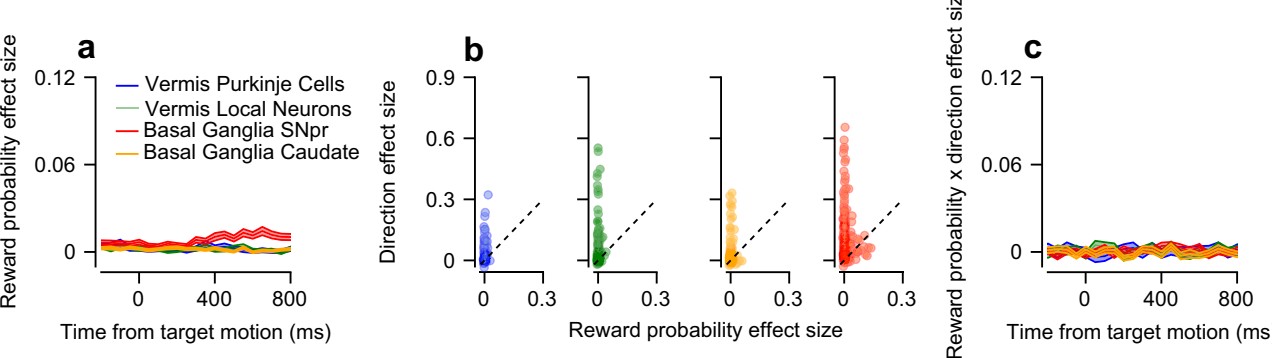

**Fig. 4 | Coding of reward probability during the motion epoch. a** Average reward probability effect size in the motion epoch. **b** Each dot represents a single neuron's reward probability (horizontal) and direction (vertical) effect sizes in the target motion epoch. Dashed lines correspond to the identity line. **c** Average reward probability x direction interaction effect size in the motion epoch.

indirect effect on neural activity. Neurons in which the reward probability condition substantially modulates the coding of direction should display high effect size values for reward probability and direction interaction (reward probability x direction). These modulations can arise from sharpening or gain modulations of the tuning curve[45]. We did not find such interactions in any of the populations (Fig. 4c), suggesting that at least on these tasks, reward and direction coding have a minimal influence on one another (Bootstrap *t*-test: $p_{SNpr} = 0.18$, $p_{caudate} = 0.41$, $p_{Purkinje} = 0.35$, $p_{local} = 0.32$). Thus, the separation in the timing of the coding of reward and movement and the lack of interaction during movement indicate that these areas multiplex reward and movement mostly across task epochs rather than simultaneously. The coding of reward during movement is thought to reflect reward bias in behavior[25]. However, our tasks were designed to minimize behavioral differences between reward probability conditions (Fig. S2) so that only a small portion of the activity in the dataset would be associated with the reward modulation of behavior.

**The coding of reward outcome is comparable in the vermis and the basal ganglia**

Reward outcome and prediction error signals were found in the basal ganglia[5] and recently in the cerebellum[1,9,46,47]. Here, we used the trials in which the reward was delivered or omitted to compare the reward outcome signals in the basal ganglia and the cerebellum directly (Fig. 5 and Figs. S9, S10). We then used the probabilistic design of the tasks to test whether the response to the reward outcome was modulated by the predictability of the reward. We pooled neurons across the pursuit and saccade tasks since in the outcome epoch, the tasks are similar.

We calculated the $\omega_p^2$ to partition the responses into reward outcome, reward probability, and direction during the outcome epoch (examples in Fig. S9). As found during movement, the reward probability effect sizes during the outcome epoch were small in comparison to the effect sizes during the cue epoch but reached significance in all populations (Fig. 5a; Bootstrap *t*-test: $p < 0.01$, in all populations). The average direction effect sizes of the population preserved the same order as when aligned to target motion (Fig. 5b; Permutation Welch's ANOVA test: $p < 0.001$, Permutation Welch's *t*-test: $p_{SNpr-caudate}$, $p_{SNpr-Purkinje}$, $p_{SNpr-local} < 0.01$). Coding of direction was expected since monkeys typically saccade back from the different eccentric positions at the end of a trial (saccade latency from target disappearance: $316 \pm 191$ ms for Monkey A, $407 \pm 107$ ms for Monkey G). Thus, finding the largest coding of movement during the outcome epoch in the SNpr supports our conclusion that the coding of eye movement direction is the most pronounced in the output of the basal ganglia. We found that the average reward outcome effect

size was slightly higher in the vermis than in the caudate and the SNpr (Fig. 5c; Permutation Welch's ANOVA test: $p < 0.001$, Permutation Welch's *t*-test: $p_{SNpr-vermis}$, $p_{caudate-vermis} < 0.001$, $p_{Purkinje-local} = 0.57$). This result differs considerably from the coding of reward probability during the cue (Fig. 2a), in which the SNpr exhibited the largest response.

We cannot completely rule out the possibility that some of the activity in this epoch was related to food consumption or other movements after reward delivery and the end of the trial. We verified that our results could not be explained by blinks and saccades that occur when the target disappeared by removing trials with blinks and equating the number of saccades across the reward outcome conditions (see Methods). We found that in these conditions, the effect sizes remained significantly positive for all populations (Bootstrap *t*-test: $p < 0.001$ for all populations), and maintained a similar order (Permutation Welch's ANOVA test: $p < 0.01$, Permutation Welch's *t*-test: $p_{SNpr-vermis}$, $p_{caudate-vermis} < 0.001$, $p_{Purkinje-local} = 0.98$).

According to the reward prediction error hypothesis, reward predictability should modulate the response to the reward outcome[2,48–50]. In terms of effect size, prediction error or any outcome modulations based on reward predictability (such as signed or unsigned prediction errors) should be expressed in a positive reward probability x reward outcome interaction effect size. However, we found only a slight interaction in all populations (Fig. 5d; Bootstrap *t*-test: $p_{SNpr}$, $p_{local} < 0.01$, $p_{Purkinje} = 0.06$, $p_{caudate} = 0.14$). The coding of reward outcome was stronger than the coding of their interaction (Fig. 5e, f; Bootstrap *t*-test: $p < 0.01$ in all populations) indicating that reward prediction error coding and any other interaction between probability and outcome were at best only weakly coded in the recorded populations. This suggests a different functional role for the recorded populations as compared to dopaminergic neurons[2] or climbing fibers[8,10] that code the reward prediction error.

In our tasks, the monkeys were not explicitly signaled when the reward was omitted, which may have caused the reward prediction signals to be spread out in time and could have impeded our ability to detect them. However, the end of each trial was clearly marked by the disappearance of the colored target, which enabled the monkeys to infer whether they would receive a reward shortly after the trial ended. In addition, we did not find differences in neural activity between rewarded trials with different reward probabilities ($P = 0.25$ reward vs. $P = 0.75$ reward), as expressed in the effect sizes for the comparison between these two conditions (not shown). In these trials, the sound of the reward pump and the availability of the reward constituted explicit indications for timing. Thus, the lack of reward prediction signals at the outcome epoch was unlikely to be the outcome as a lack of an explicit timing signal.

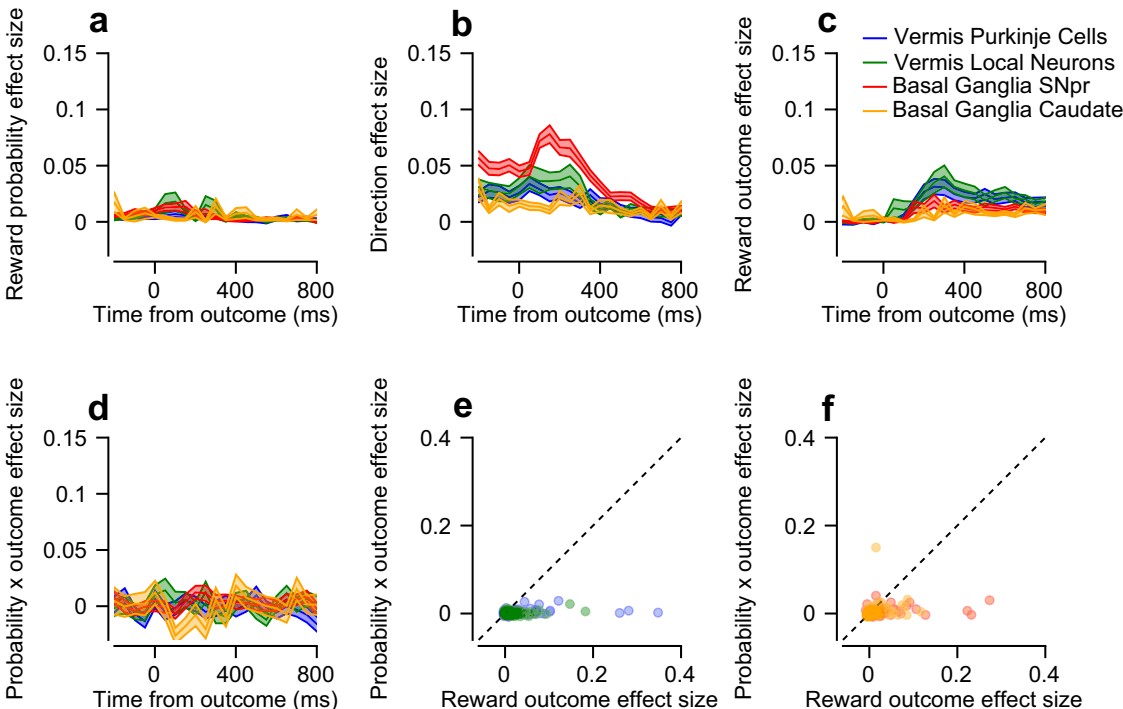

**Fig. 5 | Coding of reward outcome at the end of the trial. a–d** Reward probability (**a**), direction (**b**), reward outcome (**c**), and reward probability x reward outcome interaction (**d**) effect sizes in the outcome epoch. **e, f** Each dot represents a single neuron's reward outcome effect size (horizontal) and reward probability x reward outcome interaction effect size (vertical). **e** shows the vermis populations and **f** the basal ganglia populations.

## The coding of reward and movement in vermis complex spikes is small compared to simple spikes

Purkinje cells have the unique property of displaying two distinct types of spikes: simple and complex spikes. Simple spikes are similar to spikes observed in other neurons and were the focus of our analysis thus far. Complex spikes occur at a much lower frequency (~1 Hz, see diamonds in Fig. 1d) and are hypothesized to serve as teaching signals that affect plasticity[6,7]. Recently, complex spikes have been shown to code for reward signals[8–11]. We, therefore, extended our analysis to complex spikes recorded in the Purkinje cell population.

The complex spike reward probability effect sizes during the cue epoch (Fig. 6a; Bootstrap *t*-test: $p < 0.05$) and reward outcome during the outcome epoch (Fig. 6b; Bootstrap *t*-test: $p < 0.05$) were significant but small. During the target motion epoch, the effect size of the reward was not significantly different from 0, consistent with previous results[11] (Bootstrap *t*-test: $p = 0.55$). The coding of direction during the motion epoch on the saccade (Fig.6c; Bootstrap *t*-test: $p < 0.05$) and pursuit (Fig. 6d; Bootstrap *t*-test: $p < 0.01$) tasks was also small relative to all other populations. A paired comparison of the effect sizes of the complex and simple spikes from the same neuron indicated that for all task variables, the effect size was substantially larger for the simple spikes (scatter plots in Fig. 6a–d, Bootstrap *t*-test: $p < 0.05$ in all comparisons).

Note that effect size characterizes the signal-to-noise ratio of single neurons. Thus, a large unaccounted variability relative to the effect will result in a small effect size. We confirmed that the population response, when averaged over trials and neurons, was modulated by the direction of movement (Fig. S11). As shown above, the directional response of complex spikes was not modulated by reward expectation[11]. Thus, although complex spikes code for reward and direction, their low signal-to-noise ratio constrains their ability to transform information at the single neuron level.

## The coding of eye movement direction in the local vermis population is more pronounced in neurons with short waveforms and low firing rates

The cerebellar cortex contains multiple cell types that do not project outside the cerebellar cortex but are involved in computation. To investigate whether different cell types could represent eye movement directions to different extents, we plotted the direction effect size of local vermis neurons along with the width of their waveforms (see Methods) and their average firing rates. We found that local vermis neurons with narrow waveforms and low firing rates tended to have higher direction effect sizes (Fig. 6e, Regression: $\omega_p^2 = b_0 + b_{fr} \log(\text{firing rate}) + b_{ww} \log(\text{waveform width}) + b_{int} \log(\text{firing rate}) \cdot \log(\text{firing rate})$, $R^2 = 0.211$, $p_{model} < 0.001$, $b_0 = 1$, $b_{fr} = -0.27$, $b_{ww} = -0.16$, $b_{int} = 0.04$, $p < 0.001$ for all $b$s), indicating that specific subtypes of local neurons may code for direction to different extents. We repeated the same analysis for Purkinje cells but did not find this trend (Fig. 6F, Regression: $R^2 = 0.043$, $p_{model} = 0.4$). Previous research has suggested that the lower firing rate neurons with narrow waveforms are mossy fibers[51], implying that inputs to the vermis are more strongly tuned than interneurons or vermis output. However, since we did not find distinct clusters in our data, we refrain from making strong claims about specific populations.

## Controlling for sampling bias

One of the challenges when comparing coding across populations or structures is that the populations themselves may not be homogenous and some sub-populations may be over or under-sampled. In the current study, non-relevant neurons could have decreased the average effect size, whereas some task-relevant neurons could have been missing from our sample. This problem was overcome to some extent by focusing solely on oculomotor areas within the basal ganglia and cerebellum. Thus, we compared subsets of neurons that were very likely to participate in the task and are part of the same system. To further verify that our conclusions were not the result of sampling

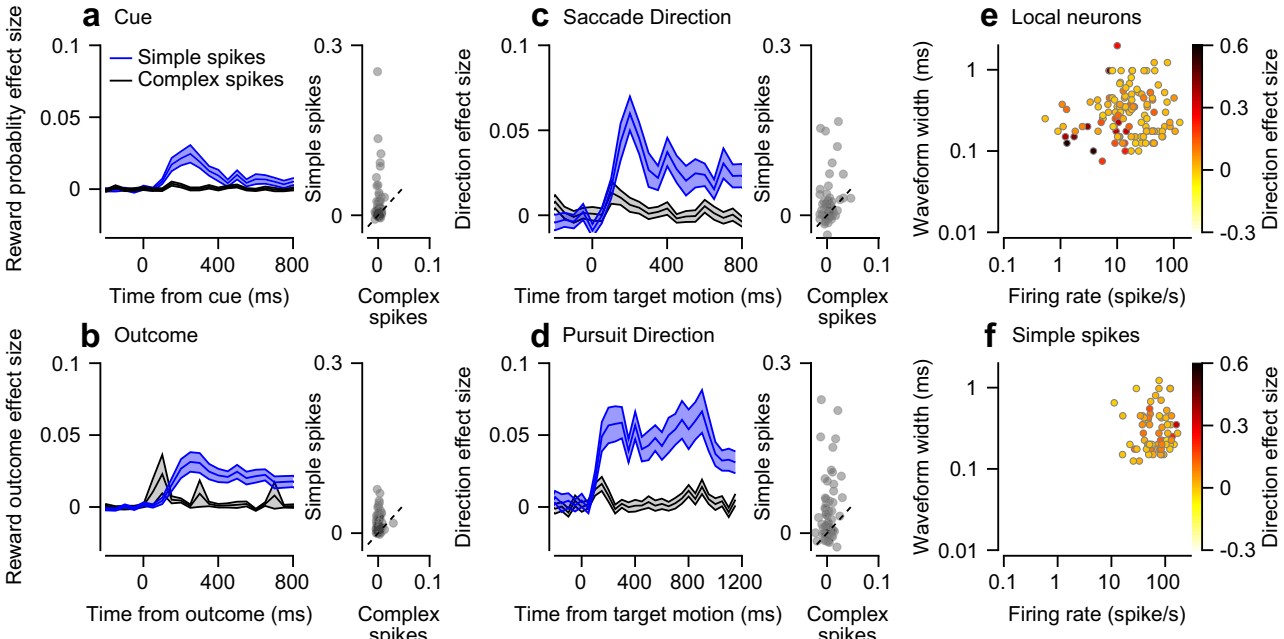

**Fig. 6 | Effect sizes of the vermis populations. a–d** Black traces represent complex spikes and blue traces show the simple spike effect sizes displayed previously. In the scatter plots, each dot represents a single Purkinje cell's complex (horizontal) and simple (vertical) spike effect sizes. The dashed lines show the identity line. **a** Reward probability effect size in the cue epoch. **b** Reward outcome effect size in the outcome epoch. **c**, **d** Direction effect size in the target motion epoch in the pursuit (**c**) and saccade (**d**) tasks. **e**, **f** Each dot represents the firing rate (horizontal) and the waveform width (vertical) of a single vermis local neuron (**e**) or a Purkinje cell's simple spikes (**f**) on a log-log scale. The color represents that neuron's direction effect size in the motion epoch of either the saccade or pursuit task.

error, we replicated the analysis solely with neurons that showed a time-varying response to the task (see Methods, Fig. S12, Table S2); namely, neurons whose firing rate was significantly different across the different time bins in an epoch.

To verify that our results were not specific to the vermis, we also extended our analysis to an additional eye movement area within the cerebellum, the flocculus complex (Fig. S13). In a previous report, we examined the coding of reward size and movement in the floccular complex and its neighboring areas[11]. We used a similar task to the pursuit task, in which the color of the cue indicated the future reward size instead of its probability (see Methods). We reanalyzed the floccular complex data and found that the reward effect sizes in the floccular complex were comparable to the vermis reward probability effect sizes. In addition, although some local floccular neurons had comparable direction effect sizes to neurons in the SNpr, the average effect size was larger in the SNpr than in the floccular complex.

## Discussion

We recorded from the basal ganglia and vermis of the same monkeys as they performed saccade and pursuit eye movement tasks in which we manipulated reward expectation. Recording from the same monkeys enabled us to control for differences between individuals and draw direct comparisons across structures. We found similar results on the saccade and pursuit tasks, which suggests that our findings are not limited to the specific parameters of our tasks, but may reflect the general properties of the eye movement system.

Our results extend recent findings on reward signals in the cerebellum[1,8–11] and relate them directly to signals found in the basal ganglia[5,24,38]. In comparison to the basal ganglia output structure, the reward expectation (Fig. 2) and sensorimotor (Fig. 3) signals in the vermis were small. Reward outcome signals in the vermis were comparable to the basal ganglia (Fig. 5), suggesting that the reward consequences of behavior are processed in both areas similarly. The findings also showed that reward outcome signals were not modulated by reward predictability (Fig. 5D), suggesting that at reward delivery,

the recorded populations are more strongly related to the processing of the current reward outcome than the prediction error.

The motor system is organized hierarchically. Neurons closer anatomically to the motor periphery show more resemblance to the final motor command[52–56]. In terms of transforming reward information into movement, a hierarchical organization of the motor system suggests that reward should be more strongly represented in the upstream areas and that motor parameters, such that eye movement direction should be more strongly represented in downstream areas. However, our results do not support this type of organization across the subcortical networks. In particular, the SNpr, which is anatomically and functionally more remote than the vermis to the motor command[18,20,23] exhibited a stronger representation of both reward expectation and movement (Figs. 2 and 3). Other inconsistencies with a hierarchical organization emerged from the comparable coding of the reward outcome for the basal ganglia and the vermis (Fig. 5).

In the basal ganglia, we found amplification of both reward and movement direction signals from the caudate to the SNpr (Figs. 2 and 3). This type of enhancement can be achieved by the convergence of noisy neurons onto a single SNpr neuron to reduce noise. Anatomically, there is a strong convergence from input to output in the basal ganglia that can support this function[57]. Thus, in the basal ganglia, noisy representations across many neurons are transformed into a smaller population with high signal-to-noise ratios. This extraction of task-relevant information is akin to computational algorithms that compress information, such as principal component analysis. Based mostly on anatomy, it has been hypothesized that the basal ganglia implement these computations[57,58]. Thus, our results provide supporting evidence for this hypothesis from the activity of neurons. We note that the response of the caudate neurons in our sample was delayed relative to the SNpr (Fig. 2a and Fig. 3a, b), implying additional inputs to the SNpr either from within the striatum or from other sources[59].

In the cerebellum, we only found a small difference in the extent to which local vermis neurons and Purkinje cells coded for reward and movement information, despite the vast convergence onto Purkinje

neurons. One possibility is that the convergence of local neurons onto Purkinje neurons may not lead to an increase in the signal-to-noise ratio. This could be due to the averaging out of opposing signals or high noise correlations limiting the cancellation of correlated noise[60–62]. While diverse inputs onto Purkinje cells may cancel each other out, they may be advantageous during the learning process by providing multiple connections that facilitate the rapid acquisition of new associations or the adjustment of movements[63]. Alternatively, the signal-to-noise ratio of local neurons and Purkinje cells might reflect recurrent connectivity within the cerebellar cortex rather than feedforward processing. For example, Purkinje cell collaterals projecting back to the cerebellar cortex[60–64] along with the projections from local neurons to the Purkinje cells may cause local neurons to code for similar information as Purkinje cells, highlighting the importance of Purkinje cell collaterals back to the cerebellar cortex[64–67]. Further research is needed to disentangle these two possibilities.

Our study focused on the oculomotor regions of the basal ganglia and cerebellum, which enabled a direct comparison of reward and movement signals in areas that contribute to the same behavior. We presented data from the input or local populations (caudate and local vermis neurons) as well as the output populations (SNpr and Purkinje cells) in both the basal ganglia and cerebellar cortex. We observed greater sharpening of the signals in the basal ganglia than in the cerebellum. However, due to their specific circuitries, defining homologies between these structures is challenging. A full comparison between the regions needs to include the entirety of the basal ganglia nuclei, the cerebellar cortex, and the deep cerebellar nuclei. In particular, the globus pallidus externus nucleus in the basal ganglia, an additional input to the SNpr, presents an interesting target for investigation, given the observed low signal-to-noise ratio in the caudate relative to the SNpr. The small effect sizes in the Purkinje cells underscore the importance of investigating the deep cerebellar nuclei. This type of exploration would help indicate whether a low signal-to-noise ratio is characteristic of the entire cerebellar network or whether there are functional distinctions between the cerebellar cortex and nuclei.

Using effect size measures, we suggested a framework to compare populations that is agnostic to the direction of the response, takes trial-by-trial variability into account, allows for comparisons of specific variables while controlling for others, and can be used to examine the interactions between variables. This framework can serve to investigate additional regions in both structures where eye movement signals have been observed[36,68] as well as in regions involved in the production of other movements. For instance, in the cerebellum, reward signals have been observed in areas associated with arm movements[69]. Similarly, in the basal ganglia, reward signals have been reported in various contexts and regions[70]. Thus, our study can form the basis for revising our understanding of the roles played by the basal ganglia and the cerebellum in the integration of reward and motor signals.

## Methods

We collected neural and behavioral data from one male (Monkey A, 8 years old, 5.5 kg) and one female (Monkey G, 7 years old, 3.7 kg) Macaca Fascicularis monkeys. All procedures were approved in advance by the Institutional Animal Care and Use Committees of the Hebrew University of Jerusalem and were in strict compliance with the National Institutes of Health Guide for the Care and Use of Laboratory Animals. We first implanted head holders to restrain the monkeys' heads in the experiments. After the monkeys recovered from surgery, they were trained to sit calmly in a primate chair (Crist Instruments) and consume liquid food rewards (baby food mixed with water and infant formula) from a tube set in front of them. We trained the monkeys to track spots of light that moved across a video monitor placed in front of them.

Visual stimuli were displayed on a monitor (55 and 63 cm from the eyes of monkeys A and G). The stimuli appeared on a dark background in a dimly lit room. A computer performed all real-time operations and controlled the sequences of target motions (Maestro, sites.google.com/a/srscicomp.com/maestro). The position of the eye was measured with a high temporal resolution camera (1 kHz, Eyelink 1000 plus, SR research) and collected for further analysis. We performed two subsequent surgeries to place a round 19 mm diameter recording cylinder over the basal ganglia and the vermis. For each structure, up to 5 quartz-insulated tungsten electrodes (impedance of 1–2 Mohm) were lowered through each cylinder using the Mini-Matrix System (Thomas Recording GmbH). When lowering the electrodes, we searched for neurons that responded during pursuit or saccade eye movements, but sometimes collected data from neurons that did not respond to eye movements.

We began by recording from the caudate and then moved to the SNpr, while continuously recording from the vermis. To record from the caudate, we approximated its depth relative to the recording chamber based on an MRI scan (Fig. S1). We then lowered electrodes towards the target. We verified that we had reached caudate by identifying typical tonically active neurons with broad waveforms and a low and regular firing rate (~5 Hz)[71]. Based on the extensive mapping of the caudate and the MRI, we estimated the location of the SNpr relative to the recording chamber. We lowered the electrodes to this location. At the targeted site, we identified neurons with a high baseline firing rate and the typical extracellular shape of SNpr neurons[72]. We confirmed that some of these neurons exhibited a clear pause during saccades in certain directions. On some recording days, we also identified neurons with pronounced eye position sensitivity and a regular firing rate, as expected from the neighboring third oculomotor nerve. To locate the oculomotor vermis, we lowered electrodes through the tentorium towards the vermis and identified areas with characteristic background responses to saccades[15]. We classified neurons as Purkinje cells by the presence of complex spikes[73,74].

Table S1 shows the number of neurons recorded from each population. We only included neurons that were recorded for at least 55 successful trials in either the saccade or the pursuit task, with a median of 141 trials. In the analyses of the cue and outcome epochs, we pooled neurons across tasks. If a neuron was recorded in more than one task, we included the task in which the neuron was recorded for more trials.

Signals were digitized at a sampling rate of 40 kHz (OmniPlex, Plexon). For the detailed data analysis, we sorted spikes offline (Plexon). For sorting, we used principal component analysis and corrected manually for errors. In some of the Purkinje cells, the complex spikes had distinct low-frequency components. In these cells, we used low-frequency features to identify and sort the complex spikes[73,74]. We paid special attention to the isolation of spikes from single neurons. We visually inspected the waveforms in the principal component space and only included neurons for further analysis when they formed distinct clusters. The sorted spikes were converted into time-stamps with a time resolution of 1 ms and were inspected again visually to check for instability and obvious sorting errors.

We also recorded licking behavior to control for behavioral differences between reward conditions that might confound the results. Licks were recorded using an infra-red beam. Monkey A tended not to extend its tongue, therefore we recorded its lip movements.

### Experimental design

Before beginning the neural recordings, we examined the behavior of the monkeys under different reward probability conditions (Fig. S2). Our selection of probabilities of 0.25 and 0.75 for recordings has several advantages. First, the considerable difference between probabilities prompted a noticeable difference in reward expectation before delivery. On an additional task in which the monkeys were

presented with two targets moving in orthogonal directions and were rewarded based on the target they tracked, the monkeys consistently chose to track the 0.75 probability target (Fig. 1c, see the Choice task section below for more details) indicating that they associated the color of the target with the probability of reward. Second, a probabilistic reward made it possible to test for differences in neural activity between reward delivery and omission (reward outcome) and manipulate reward expectation at delivery to test whether the neural activity was consistent with the reward prediction error hypothesis[48]. Third, the effect of the reward probability on pursuit and saccades was small in comparison to a condition in which the monkeys never received a reward ($P = 0$, Fig. S2), and in comparison, to the variability in the behavior within conditions. This similarity in behavior served to examine the responses to reward information separately from movement preparation and execution.

**Pursuit task.** Each trial started with a bright white target that appeared in the center of the screen (Fig. 1a). After 500 ms of presentation, in which the monkey was required to acquire fixation, a colored target replaced the fixation target. The color of the target signaled the probability of receiving a reward upon successful tracking of the target. For monkey A, we used yellow to indicate a 0.75 probability for reward and green to indicate 0.25. For monkey G, we reversed the associations. After a variable delay of 800–1200 ms, the targets stepped in one of eight directions (0°, 45°, 90°, 135°, 180°, 225°, 270°, 315°) and then continuously moved in the opposite direction (step-ramp)[30]. For both monkeys, we used a target motion of 20°/s and a step to a position 4° from the center of the screen. The target moved for 750 ms for monkey A and 650 ms for monkey G and then stopped and stayed still for an additional 500–700 ms. If the monkey's gaze was within a 3–5° × 3–5° window around the target, the monkey received a reward according to the probability specified by the color.

**Saccade task.** The structure of the saccade task was identical to the pursuit task, except for the target motion epoch. In the saccade task, following the random delay, the central colored target disappeared and immediately reappeared in one of 8 eccentric locations (0°, 45°, 90°, 135°, 180°, 225°, 270°, 315°) 10° from the center of the screen. If the monkey's gaze was within a 5° × 5° window around the target, the monkey received a reward according to the probability specified by the color.

**Choice task.** We used this task to verify that the monkeys associated the color of the target with the reward probability. The structure of the task was similar to previous reports[75]. The monkeys were required to choose one of two targets with different reward probability values presented on the screen. Their choice determined the probability of receiving a reward. Each trial began with a 500 ms fixation period, similar to the tasks described previously. Then, two additional colored spots appeared at a location eccentric to the fixation target. One of the colored targets appeared 4° below or above the fixation target (vertical axis) and the other appeared 4° to the right or left of the fixation target (horizontal axis). We use the same colors as we used in the saccade and pursuit tasks. The monkeys were required to continue fixating on the fixation target in the middle of the screen. After a variable delay of 800–1200 ms, the white target disappeared, and the colored targets started to move towards the center of the screen (vertically or horizontally) at a constant velocity of 20°/s. After a variable delay, the monkeys typically made saccades toward one of the targets. We defined these saccades as an eye velocity that exceeded 80°/s. The target that was closer to the endpoint of the saccade remained in motion for up to 750 ms and the more distant target disappeared. The monkeys were required to track the target until the end of the trial. After the end of the motion, the remaining target stayed still for an additional 500–700 ms, and then the monkeys received a reward

according to the probability determined by the color of the chosen target. To allow the monkeys to choose the pursuit target freely, we suspended fixation requirements from the time of disappearance of the fixation target to the end of the first saccade. After the saccade, the monkeys' gaze had to stay within a 4–5° × 4–5° window around the target. Upon successful performance, the monkeys received a reward with the probability defined by the color of the target it tracked.

**Reward size task.** We used previously analyzed data recorded from the floccular complex and adjacent areas, while the monkeys performed a smooth pursuit task in which we manipulated reward size. The full details can be found in previous reports[11,76]. Briefly, the temporal and target motion properties of the task were the same as the reward probability pursuit task described above. However, in this task, the color of the target signaled the size of the reward the monkeys would receive if they tracked the target. One color was associated with a large reward (-0.2 ml) and the other with a small reward (-0.05 ml).

### Data analysis
All analyses were performed using MATLAB (MathWorks).

**Eye movement analysis.** To calculate the average of the smooth pursuit initiation, we first removed the saccades and treated them as missing data. We used eye velocity and acceleration thresholds to detect saccades automatically and then verified the automatic detection by visual inspection of the traces. The velocity and acceleration signals were obtained by digitally differentiating the position signal after we smoothed it with a Gaussian filter with a standard deviation of 5 ms. Saccades were defined as an eye acceleration exceeding 1000°/s², an eye velocity crossing 15°/s during fixation, or an eye velocity crossing 50°/s while the target moved on the pursuit task. We then averaged the traces in the target movement direction. Finally, we smoothed the traces using a Gaussian filter with a standard deviation of 20 ms.

**Peristimulus time histogram.** To examine the average time-varying properties of the response, we calculated the peristimulus time histogram (PSTH) at a 1 ms resolution. We then smoothed the PSTH with a 20 ms standard deviation Gaussian window, removing at least 100 ms before and after the displayed time interval to avoid edge effects.

**Tuning curves.** To calculate the tuning curves (Figs. S8, S11), we averaged the responses in a time window relative to the target motion onset. In Fig. S11, we focused on the first 100–300 ms of the movement, based on the time window in which we previously observed complex spike response to visual motion and eye movements[11]. In Fig. S8, we used the entire motion epoch. We calculated the preferred direction of the neuron as the direction that was the closest to the vector average of the responses across directions (direction of the center of mass). We used the preferred direction to calculate the population tuning curve by aligning all the responses to the preferred direction.

**Waveform width.** The waveform width was calculated by first aligning the waveforms to their minimal points, and then calculating the time difference between the peak and the trough. The waveforms were bandpass filtered online with a cut-off frequencies of 0.034 Hz and 8 kHz, and bandpass filtered again off-line with cut-off frequencies of 250 Hz and 6 kHz.

**Effect size in time.** To quantify a neuron's coding of a variable over time, we calculated the $\omega_p^2$ effect size in each 50 ms bin separately. In each bin and each trial, we counted the number of spikes, resulting in a distribution of the number of spikes emitted by the neuron in each

condition. $\omega_p^2$ is a common effect size measure used in ANOVA designs that is often preferred over other effect size measures since it is unbiased[35,77,78]. The calculation is as follows:

$$\omega_p^2 \equiv \frac{SS_{effect} - \frac{df_{effect}}{df_{error}} \cdot SS_{error}}{SS_{effect} + \frac{(N-df_{effect})}{df_{error}} \cdot SS_{error}} \qquad (1)$$

where $SS_{effect}$ is the ANOVA sum of squares for the effect of a specific variable, $SS_{error}$ is the sum of squares of the errors after accounting for all experimental variables, $df_{effect}$ and $df_{error}$ are the degrees of freedom for the variable and the error, respectively, and $N$ is the number of observations (number of trials). During the cue epoch, the reward probability was the only variable in the ANOVA design, during the target motion epoch the variables were the reward probability and direction, and during the outcome epoch, the variables were rewarded probability, direction, and the reward outcome. We also included all possible interactions. We used the Type II sum of squares to account for unequal group sizes (for example, a smaller number of trials in the reward-delivered as compared to the reward-omitted within the $P = 0.25$ condition)[79]. We used the partial effect size since it enables a better comparison between neurons that respond or do not respond to other experimental variables. Using the full effect size did not alter any of our conclusions.

**Effect size in the entire epoch.** To calculate the coding of a variable throughout the entire epoch, we calculated $\omega_p^2$ in models that included time as an additional variable. Again, we calculated the number of spikes in 50 ms bins, 0 to 800 ms after an event during the trial (cue appearance, target motion, or outcome). Changing the bin size yielded highly correlated effect sizes (Fig. S4d–i). We fitted an ANOVA model that included the same variables as listed above for each epoch, with the addition of time (the specific bin the sample came from). Again, we included all possible interactions. The formula that we used was similar:

$$\omega_p^2 \equiv \frac{SS_{effect} + SS_{effect\,x\,time} - \frac{df_{effect} + df_{effect\,x\,time}}{df_{error}} \cdot SS_{error}}{SS_{effect} + SS_{effect\,x\,time} + \frac{(N - df_{effect} - df_{effect\,x\,time})}{df_{error}} \cdot SS_{error}} \qquad (2)$$

where $SS_{effect\,x\,time}$ is the ANOVA type II sum of squares for the interaction of a specific variable with time, and $df_{effect\,x\,time}$ are the corresponding degrees of freedom. We included the interaction term since it quantifies the time-varying coding of the variable. In this case, $N$ is equal to the number of trials x the number of time bins.

We verified that $\omega_p^2$ was not determined by the firing rate of neurons (Fig. S5). Although Purkinje cells had the highest firing rates, their effect sizes during the cue and target motion epochs were relatively small. Additionally, in some cases, the rate and effect sizes were correlated, but this correlation was small or absent in others.

**Statistical tests.** The distributions of effect sizes tended to be non-normal and the variances were different between populations. We, therefore, avoided using standard ANOVA tests. Instead, we used permutation and bootstrap methods to evaluate the distributions of statistics that correct for unequal variances. When permuting we recalculated the statistics 10,000 times, since we observed that the p-values were stable at this number of repetitions.

*Permutation one-way Welch's ANOVA test:* We permuted the population labels of the effect sizes and calculated the Welch's ANOVA F statistic for each permutation (https://www.mathworks.com/matlabcentral/fileexchange/61661-wanova, version 1.0). The p-value was defined as the probability for the F statistic to be larger than the F statistic for the real sample.

*Permutation Welch's t-test:* We permuted the population labels of the effect sizes and calculated the Welch's t-test statistic for each permutation. The p-value was defined as the probability for the t statistic to be more extreme than the t statistic for the real sample (two-tailed test).

*Bootstrap t-test:* We used this to test the null hypothesis that effect size values, or the paired differences in effect size values, are significantly different than 0. To simulate the null hypothesis, we subtracted the sample mean from the sample. We then resampled (with repetition) from our the mean subtracted sample. The p-value was defined as the probability for the t statistic to be farther than 0 from the t statistic of the real sample (two-tailed test).

*Significance of single neurons:* To test whether an individual neuron differentiates between the conditions of an experimental variable (Table 1), we used the same ANOVA designs described above in the Effect Size subsection, including time as a variable. We tested the combined significance of the main effect of a variable and its interaction with time, by summing the sum of squared and the degrees of freedom.

**Time-varying response.** We repeated our analysis on subsets of neurons within each population that changed their average firing rates significantly throughout the epoch (Fig. S12, Table S2). To find these neurons, we calculated the significance of the time variable, using the same ANOVA design described above in the Effect size in the Entire epoch subsection.

**Comparison to a classifier.** We compared the reward probability effect size to the 10-fold cross-validated accuracy of a classifier for the reward condition (Fig. S4). To train the classifier, we calculated the PSTH of the training set for each reward condition. To predict the label of each trial in the test set, we calculated its single trial smoothed PSTH. We compared the PSTH to the PSTHs calculated over the training set and assigned it the label of the PSTH most similar to it (minimal $L_2$ norm).

**Blink and saccade control during the outcome epoch.** Monkeys tended to blink at the end of trials, immediately after the target disappeared and the reward had either been delivered or omitted. The blink frequency was slightly higher at the end of trials when the reward was omitted than when it was delivered (a difference of approximately 3 Hz). The monkeys also tended to saccade back towards the center of the screen at the end of trials, with a slightly higher frequency of early saccades when a reward was delivered than when it was omitted (a difference of approximately 2 Hz). To confirm that the reward outcome effect size did not result from differences in the pattern of saccades or blinks across task conditions, we repeated the analysis after removing trials with a blink in the first 500 ms, and equated the number of saccades between reward conditions by randomly discarding trials. Even after this manipulation, the effect sizes were significantly positive in all populations. This suggests that the reward outcome effect persisted regardless of the occurrence of blinks and early saccades.

## Reporting summary

Further information on research design is available in the Nature Portfolio Reporting Summary linked to this article.

## Acknowledgements

This work is dedicated to the memory of Mrs. Lily Safra, a great supporter of brain research. We thank Lottem E. and Fox L. for commenting on early versions of the manuscript. This project received funding from the European Research Council (ERC) under the European Union's Horizon 2020 research and innovation program (grant agreement No 755745, M.J.) and the Israel Science Foundation (380/17, M.J.).

## Author contributions

The experiment was designed by all three authors. The data were collected by NL and GZ, who recorded from the vermis and basal ganglia, respectively. The data were analyzed by NL and GZ. The manuscript was written by NL and MJ, and all authors were involved in the interpretation of the results.

## Competing interests

The authors declare no competing interests.

## Data availability

The entirety of the data generated in this study has been deposited in the Dryad database, https://doi.org/10.5061/dryad.73n5tb33r[80]. Source data are provided with this paper.

## Code availability

All code used in this study is available on GitHub (https://github.com/noga-larry)[81].

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
