## [Peer Review File · Nature Communications]

Organization of reward and movement signals in the basal ganglia and cerebellumREVIEWER COMMENTS

Reviewer #1 (Remarks to the Author):

The basal ganglia have been known to control motor actions especially using reward-related information. Indeed, previous studies have shown that neurons in the basal ganglia exhibit neuronal modulations evoked not only by motor actions but also by rewards. By contrast, the cerebellum has been hypothesized to be involved in motor control and error-based motor learning. However, recent studies have reported that neurons in the cerebellum also encode reward-related signals in a manner similar to the basal ganglia, suggesting that both structures are involved in reward-based motor control. The fundamental difference between these structures now becomes controversial.

To tackle this issue, the present study compared neuronal signals in the basal ganglia and cerebellum by recording from their oculomotor areas in the same tasks and same monkeys. The monkeys performed oculomotor tasks in which a sensory cue predicted the probability of reward. The authors recorded neuronal activity from the input and output structures of the basal ganglia (caudate and SNpr, respectively) and the vermis of the cerebellum. Because the cerebellum is anatomically closer than the basal ganglia to the motor neurons that control eye movements, the authors initially expected that the reward signals should be more pronounced in the basal ganglia than in the cerebellum and that movement signals should be more pronounced in the cerebellum than in the basal ganglia. Similarly, they expected that reward signals should be more pronounced in the input structure of the basal ganglia (caudate) than in the output structure (SNpr) and that movement signals should be more pronounced in the output structure than in the input structure.

In contrast with their initial assumption, the authors found that not only movement signals but also reward signals were most pronounced in the output structure of the basal ganglia (SNpr). The caudate and cerebellum vermis exhibited weaker movement and reward signals compared with the SNpr. This is their main finding and really surprising and novel. However, I think this finding can be explained by a sampling bias as I will mention later. Anyway, the aim of this study is timely because many recent studies in the field focus on roles of cognitive and motivational cerebellar signals in behavior. I would like to see the responses of the authors to the following my comments before final decision.

Major:

(1) The authors mentioned “When lowering the electrodes, we searched for neurons that responded during pursuit or saccade eye movements, but sometimes collected data from neurons that did not respond to eye movements.” in STAR Methods. This means they recorded mainly from motor-related (i.e., eye movement-related) neurons. Although the authors recorded from eye movement-related areas in the basal ganglia and cerebellum, many neurons in the areas do not encode eye-movement signals

but encode reward-related signals especially in the input structure of the basal ganglia (caudate). See table 3 of Kawagoe, Takikawa & Hikosaka (2004) J Neurophysiol. They found the proportion of reward selective neurons is larger than that of saccade-direction selective neurons in the caudate, suggesting that many caudate neurons are selective only to reward. Therefore, the present study seems to miss “pure reward selective neurons”. Neurons that are purely sensitive to reward are likely to more distribute in areas anatomically farther from motor neurons. That is, the caudate contains more pure reward sensitive neurons than the SNpr and vermis in the cerebellum. Because of the sampling bias, the authors seem to miss these pure reward sensitive neurons and are likely to underestimate the effect size of reward probability in the input structure of the basal ganglia (caudate) and/or to overestimate the effect size in the SNpr. This is a serious problem in this paper and needs to be solved. Although the authors mentioned “controlling for sampling bias” in the result section and Figure S8 by analyzing neurons that showed “a time-varying response” to the task, they did not explain what the time-varying response is in the manuscript. So, I cannot judge whether the sampling bias problem is solved.

(2) The authors used $\omega p2$ to measure the effect sizes of reward probability and eye movement direction on neuronal activity, which is not familiar in the field at least for me. Researchers in the field of systems neuroscience often use “area under Receiver Operating Characteristic (ROC) curve” to evaluate to what degree neuron activity detects a signal in noise. The authors mentioned the reasons why they used $\omega p2$ in the manuscript. The firstly written reason is that $\omega p2$ is agnostic to the direction of modulation. That is, this index represents the magnitude of signal but not its direction. For example, some neurons have higher firing rate when reward probability was high, while others have higher firing rate when reward probability was low. This index does not reflect such modulation direction. However, the direction of modulation is very important. Indeed, neurons with different modulation directions sometimes show distinct electrophysiological properties. I strongly recommend the authors to use the well-accepted index, area under ROC curve. Otherwise, skeptic readers don’t believe their results.

Minor:

(1) The authors show only $\omega p2$ for population analyses. I would like to see population (averaged) PSTHs of each type of neurons, which are closer to raw data, because the profile of $\omega p2$ data seems to be different from that of conventional electrophysiological data. For example, it has been well known that many neurons in the basal ganglia including the caudate respond to outcome. But, in the present study, the response of caudate neurons to reward outcome, which is shown as $\omega p2$, is tiny. Readers including me may become suspicious about the quality of data.

(2) I would like to see the latencies of the reward- and eye movement-related signals in the caudate, SNpr, and the vermis in the cerebellum. The differences of these signal latencies may provide important insights to understand functional segregation between the basal ganglia and cerebellum.

(3) The results obtained by the convergence model is described in the discussion section, not in the result section. I feel the description is short and not enough to fully understand the model and results. I recommend the authors to add new sub-section in the result section to explain details of the model and results.

Reviewer #2 (Remarks to the Author):

In this paper, Larry and colleagues study the representation of reward and movement signals in the cerebellar vermis and the basal ganglia in awake behaving primates. The authors conclude that both these structures have different organization for reward and movement representations. This is an important area of research because 1) it adds more evidence for reward processing in the cerebellum and tries to elucidate mechanisms of reward processing that are currently poorly studied and 2) more importantly, the paper studies 2 brain areas and compare their properties in terms of reward and movement which further sheds light on circuit level properties. The experiments build on previously published task and have similar experimental design and set up. However, I have several conceptual and analysis based concerns that significantly dampen my enthusiasm about the work. This is laid out below. This work needs significant revision to be considered for publication in my opinion.

Major points:

1. My biggest concern about this paper is the disconnect between the question asked and area of the Cerebellum studied. I really doubt if the vermis is the right place to study the questions the authors pose. Although vermis is undoubtedly involved in saccade and pursuit, as the authors noted in this and their previous paper (Lixenberg et al., 2020), the vermis and the flocculus are more tightly connected with the motor system than the hemispheres and thus don't represent reward in their signal as much, simply because of the anatomy. In fact, the hemispheres, for example crus I and II are monosynaptically connected with areas of basal ganglia that process reward (Bostan et al 2010). And Purkinje cells in areas within crus I and II are also shown to process reward (Sendhilnathan et al., 2020 and 2022 - although these are hand related Purkinje cells). In the light of such recent literature, areas within the hemispheres that drive eye movements seem to be a more suitable candidate for the questions the authors ask here. However, if the question were not 'which cerebellar area processes reward in conjunction with the basal ganglia' but rather 'does vermis process reward', then the authors' approach seems more reasonable. Even then, studying a part of the cerebellum (vermis) that is not predominantly involved in reward processing due to its lack of anatomical connections with reward processing areas of the brain including basal ganglia and concluding that it does not represent reward as much, seems very unsurprising and lacks novelty.

2. Please substitute "cerebellum" with the precise location studied (i.e vermis) throughout the paper. Concluding that 'cerebellum does not represent reward expectation as much' is completely unwarranted

since vermis is not representative of the entire cerebellum. This is as if one is recording from V1 and concluding that the 'cortex does not represent faces'. Recording from IT and not V1 will enable researchers to study face representations – similarly recording from an appropriate area of the cerebellum would enable studying reward representations in the cerebellum.

3. Where exactly were the recording made in the vermis and basal ganglia? There's no description or MRI or electrode tract tracing of where the recording was made from the vermis or basal ganglia. This makes the interpretation of results a bit challenging. Furthermore, were neurons from basal ganglia and the cerebellum simultaneously recorded? This wasn't clear anywhere.

4. I don't see the utility of the convergence model. I am not sure what new information one learns from it. As I understand it, it says as the input signal gets more homogenous, the output signal gets "sharper". But isn't this trivial? More heterogenous input would lead to more averaged output signal and thus, less sharp signal. Furthermore, several physiological aspects of this model are not clear. First, the blue and red neurons are hypothesized to be neurons that respond to .75 or .25 reward probability. However, did authors really find these neurons in their ephys explorations? The only example comes from fig 1 but those neurons don't have impressive differences except for 1F - especially since the spike density functions lack error bar, it's hard to tell if the differences are actually significant or not. Second, the model by Herzfeld et al 2015 suggests that only P-cells with similar properties (CS direction) converge to the same nucleus neuron and this best explains saccade error direction coding in cerebellum. The model in the current manuscript seems to be a generalized version of the Herzfeld model but I am not convinced how the heterogeneous aspect will work out due to "less sharp" signals. Third, and related to the above point, what is the relationship between CS direction and the blue and red neurons here? Apart from this, what is the motivation for the model? Is it to explain the ephys findings or to make any prediction? I am not sure if it does either. In short, I don't think I learn much from the model and it raised more questions than provide answers to me.

Minor points:

1. In line 99-100 the authors mention they identify neurons that are reward selective. How was this done? By eyeballing or using any statistical test? Please describe this method.
2. Does the effect size depend on both 1) difference in the value of the variable for a given condition (for example .75 reward) relative to noise and 2) the trial-to-trial difference in the activity within a condition? In that case, if the effect size is low, it could be because of either or both of the above reasons. How do the authors disambiguate it?
3. Related to the point above, since SNR for CS is low, and since they only fire in about 30-40% trials, the trial-to-trial variability is also high and therefore I highly doubt if the effect size analysis is the right way to analyse the CS data. Furthermore what exactly is the conclusion of the CS analysis?

4. What are “local neurons”? Golgi, interneurons etc? Please define them properly since different classes of neurons in the cerebellar microcircuit have different spike waveforms, inter spike interval distributions and different functions.

5. line 288 typo - reanalysis

6. Lines 206-207 although the neurons are tuned to saccade direction as well as pursuit directions, is the coding direction the same for saccade and pursuit? This is not clear.

7. Are the plots in Fig S6 only for correct trials? Since trial outcome and reward outcome can be disassociated in this task, I wonder how the neural activity would look for these two different parameters.

8. The discussion feels insufficient. Greater insights can be drawn from the results. Currently the discussion is heavily dominated by the model which doesn't seem that useful in my opinion (see major point 4). The authors record from two brain areas here – what can be said not just about these areas individually but the relationship between them at the level of a circuit?

Reviewer #3 (Remarks to the Author):

This study in behaving macaques aimed to compare activity of neurons in the basal ganglia and cerebellum of the same monkeys using the same tasks, thus providing consistency across the neural samples that helps to interpret the relative functions of the areas. The tasks varied probability of receiving reward for correct responses using an instructional cue, as well as direction of saccadic and smooth pursuit eye movements. The study is presented in a framework that purports to study the inputs and outputs of both the basal ganglia and cerebellum, so as to facilitate understanding of the signal transformations within each subcortical network. The key hypotheses are that reward signals are more prevalent at the inputs while movement direction signals are more prevalent at the outputs, and that the cerebellum has more of a movement role than the basal ganglia. Large samples of neurons were recorded in the caudate and substantia nigra pars reticulata (input and output of basal ganglia, respectively) and non-Purkinje cells and Purkinje cells of the cerebellar cortex (input and output of cerebellum, respectively). In general, the hypotheses were refuted in ways that are potentially very interesting and highlight newly appreciated differences between basal ganglia and cerebellar processing.

Overall, the manuscript provides a wealth of data that will be interesting to experts. The data are collected and analyzed rigorously, for the most part. The framework as described above is interesting, and if tested specifically to compare analogous processing zones of the basal ganglia and cerebellum, could yield compelling results. The choice of basal ganglia recording regions (caudate and SNr) conform to the framework well, but the choice of cerebellar cortical neurons do not. The report presents a worthy dataset but I have concerns that the weakness in matching regions of study to the conceptual input-output framework may limit the general impact of the work.

Big picture comments:

1. The main cerebellar neurons studied were Purkinje cells. These are neither at the input nor the output of the cerebellum. The paper positions them at the output. This is true for cerebellar cortex, but not for cerebellum as a whole. Purkinje cells innervate neurons of the deep cerebellar nuclei (and some other targets), that are considered to provide cerebellar output to the rest of the brain.

Moreover, the paper positions the non-Purkinje cell neurons in cerebellar cortex as comprising an input stage. First, they are not the input stage; it is generally accepted that granule cells are. But they are not even necessarily open-loop inputs to the Purkinje cells. Collaterals from Purkinje cell axons are known to feed back to and modulate the cerebellar cortex interneurons. A few references:

Crook, J.D., Hendrickson, A., Erickson, A., Possin, D. and Robinson, F.R., 2007. Purkinje cell axon collaterals terminate on Cat-301+ neurons in Macaca monkey cerebellum. *Neuroscience*, 149(4), pp.834-844.

Hirono, M., Saitow, F., Kudo, M., Suzuki, H., Yanagawa, Y., Yamada, M., Nagao, S., Konishi, S. and Obata, K., 2012. Cerebellar globular cells receive monoaminergic excitation and monosynaptic inhibition from Purkinje cells. *PLoS one*, 7(1), p.e29663.

Guo, C., Rudolph, S., Neuwirth, M. E., & Regehr, W. G. (2021). Purkinje cell outputs selectively inhibit a subset of unipolar brush cells in the input layer of the cerebellar cortex. *Elife*, 10, e68802.

Witter, L., Rudolph, S., Pressler, R. T., Lahlaf, S. I., & Regehr, W. G. (2016). Purkinje cell collaterals enable output signals from the cerebellar cortex to feed back to Purkinje cells and interneurons. *Neuron*, 91(2), 312-319.

In other words, the cerebellar cortical interneurons and Purkinje cells do not necessarily mediate a serial flow of information, but may form a closed loop microcircuit. This is not addressed.

I recognize that achieving the ideal match of input and output stages between basal ganglia and cerebellum would be exceedingly difficult in monkeys, primarily because the granule cells of cerebellum (the input stage there) are technically challenging to study. But it does not seem infeasible to match the output stages better, i.e. SNr and a deep cerebellar nucleus. Put simply, the framework did not seem to drive the selection of recording sites; it appears more that an impressive data set, across multiple areas, was collected and then a framework was constructed to justify direct comparisons across the areas. But due to the mismatch of the cerebellar sites to the input-output concept, that framework is not really appropriate.

I do emphasize, however, that this is a beautiful data set. It will be of great interest to basal ganglia and cerebellar researchers. It may be more straightforward to present the data more plainly, with a more scholarly discussion of what is learned from the comparisons that can be made and the limitations of the comparisons.

2. The results in basal ganglia not only refute the hypothesis that neural signals transition from more reward-related to more movement-related from caudate to SNr (which is interesting), but also throw a paradoxical monkey wrench into the framework of caudate as input and SNr as output. The caudate signals are not only weaker, they are later than the SNr signals. As pointed out by the authors, this could mean that the SNr is receiving substantial inputs of non-caudate origin. That is also very interesting, but if the framework breaks down for basal ganglia, and did not seem to motivate the selection of recording sites in the cerebellum, it is unclear how the use of the framework promotes clear communication of the results.

Major data comments:

1. It is interesting that no region appears to encode reward prediction error (RPE) strongly, although, as the authors note, the study does not appear to be specifically designed for this, as this epoch is confounded with sipping. Furthermore, because the RPE would occur at reward delivery, the animal no longer needs to fixate, so it is unclear what the eyes are doing in this period (perhaps staring at the juice tube). With the current study design, it is unclear when the animal should experience a negative RPE in the case of reward omission (and may end up smearing any signal across time) because the animal must infer the omission through a timing task – comparing a noisy estimate of how much time has passed since task completion to a noisy expectation of when the reward should arrive by if delivered. I don't think one can speak to coding of reward outcome with this task design (Fig 4). The eye could be doing anything in this phase, so a saccade from the target to the juice tube could account for directional encoding at this epoch (Fig 4B) and differences between what the eyes do for reward delivery/omission could account for the reward outcome effect (Fig 4C). If the authors can account for condition-dependent eye movements/position during this epoch, that could address this problem.

2. In the saccade task the target always appeared at 10 degrees. The authors don't mention the receptive/movement field eccentricity or size for the regions and neurons they recorded from. With the saccade targets always at 10 degrees, the measure of directional coding would seem to insufficiently characterize neurons that may have relatively punctate, diagonally aligned, or smaller eccentricity regions of sensitivity. This suggests that differences in response field locations or sizes across the recording sites may account for some of the differences between cerebellum/caudate vs. SNr directional encoding.

3. While omega squared seems to be a reasonable choice for quantifying reward signals, in the case of directional encoding, the authors should provide some explanation to justify its use. There are many ways neurons can encode movement beyond directional sensitivities. Measures of timing and vigor, for example, may contribute additional useful information for comparing across areas.

Minor comments:

1. The manuscript is missing some important behavioral measures. How did performance (in terms of % correct) in the 25% reward probability condition compare to the 75% reward probability condition? Did animals have any directional biases in terms of performance and reaction time? Different reaction times across different target/target motion directions could create artificial differences in how neurons respond to each direction. This is primarily a problem because Fig 3A/B are aligned to target motion, not saccade/pursuit onset.

2. Fig 1C task details are lacking. Did the monkey select the stimulus via a saccade? Where were the stimuli placed? A brief description would be helpful here.

3. Fig S1 - The authors find significant differences between behavior in the two reward conditions, but never discuss whether or not this is important in the interpretation of their results. The differences are not an issue in Fig 1A which is before movement, but could impact the interpretation of Fig 1B, since these small differences in movement metrics could lead to spillover from movement coding into reward coding.

4. Line 146: pSNpr-purkinle -> purkinje

5. Lines 224-225: What is this saccade latency relative to – reward, or disappearance of the target?

6. Fig 1: No error bands, so it's difficult to tell what is noise. Black diamonds are in D not F (legend line 11).

7. Fig 3: It would be helpful to show some measure of correlation for plots like C, otherwise I don't know what to make of these plots.

8. The model may be better placed in the Results, not the Discussion. It seems to be more than just a "Discussion figure" that provides a take-home message; it involved careful choice of parameters, selection of model approach, etc, that feels out of place as a concluding point.

Reviewer #1 (Remarks to the Author):

The basal ganglia have been known to control motor actions especially using reward-related information. Indeed, previous studies have shown that neurons in the basal ganglia exhibit neuronal modulations evoked not only by motor actions but also by rewards. By contrast, the cerebellum has been hypothesized to be involved in motor control and error-based motor learning. However, recent studies have reported that neurons in the cerebellum also encode reward-related signals in a manner similar to the basal ganglia, suggesting that both structures are involved in reward-based motor control. The fundamental difference between these structures now becomes controversial.

To tackle this issue, the present study compared neuronal signals in the basal ganglia and cerebellum by recording from their oculomotor areas in the same tasks and same monkeys. The monkeys performed oculomotor tasks in which a sensory cue predicted the probability of reward. The authors recorded neuronal activity from the input and output structures of the basal ganglia (caudate and SNpr, respectively) and the vermis of the cerebellum. Because the cerebellum is anatomically closer than the basal ganglia to the motor neurons that control eye movements, the authors initially expected that the reward signals should be more pronounced in the basal ganglia than in the cerebellum and that movement signals should be more pronounced in the cerebellum than in the basal ganglia. Similarly, they expected that reward signals should be more pronounced in the input structure of the basal ganglia (caudate) than in the output structure (SNpr) and that movement signals should be more pronounced in the output structure than in the input structure.

In contrast with their initial assumption, the authors found that not only movement signals but also reward signals were most pronounced in the output structure of the basal ganglia (SNpr). The caudate and cerebellum vermis exhibited weaker movement and reward signals compared with the SNpr. This is their main finding and really surprising and novel. However, I think this finding can be explained by a sampling bias as I will mention later. Anyway, the aim of this study is timely because many recent studies in the field focus on roles of cognitive and motivational cerebellar signals in behavior. I would like to see the responses of the authors to the following my comments before final decision.

Major:

1. The authors mentioned “When lowering the electrodes, we searched for neurons that responded during pursuit or saccade eye movements, but sometimes collected data from neurons that did not respond to eye movements.” in STAR Methods. This means they recorded mainly from motor-related (i.e., eye movement-related) neurons. Although the authors recorded from eye movement-related areas in the basal ganglia and cerebellum, many neurons in the areas do not encode eye-movement signals but

encode reward-related signals especially in the input structure of the basal ganglia (caudate). See table 3 of Kawagoe, Takikawa & Hikosaka (2004) J Neurophysiol. They found the proportion of reward selective neurons is larger than that of saccade-direction selective neurons in the caudate, suggesting that many caudate neurons are selective only to reward. Therefore, the present study seems to miss “pure reward selective neurons”. Neurons that are purely sensitive to reward are likely to more distribute in areas anatomically farther from motor neurons. That is, the caudate contains more pure reward sensitive neurons than the SNpr and vermis in the cerebellum. Because of the sampling bias, the authors seem to miss these pure reward sensitive neurons and are likely to underestimate the effect size of reward probability in the input structure of the basal ganglia (caudate) and/or to overestimate the effect size in the SNpr. This is a serious problem in this paper and needs to be solved. Although the authors mentioned “controlling for sampling bias” in the result section and Figure S8 by analyzing neurons that showed “a time-varying response” to the task, they did not explain what the time-varying response is in the manuscript. So, I cannot judge whether the sampling bias problem is solved.

We agree with the reviewer's comment that without a complete sampling of a structure, it is possible that signals in unsampled areas could be different. Nevertheless, since a comparison between areas is critical for understanding how the brain operates at the system level, it is important to study how different areas process information. We believe that our study provides a critical and missing comparison between the basal ganglia and cerebellum.

We focused on the oculomotor areas to compare areas while minimizing the limitations of comparative studies. By examining the populations of neurons that are involved in the production of the same behaviors, we can compare reward and movement signals more directly across structures. Specifically, after identifying the oculomotor areas, we did not limit ourselves to eye movement neurons. Therefore, the movement and reward signals reported in this manuscript reflect the distribution of the signal within the oculomotor areas. In the revised manuscript, we have highlighted these important points. We also included a discussion on the coding of reward in additional regions of the basal ganglia and the cerebellum and the limitations of the comparative approach.

We have added a clarification in the Methods section of the revised manuscript on what we mean by "time-varying response". We are referring to the subset of neurons whose activity is modulated in time across all conditions.

2. The authors used $\omega p2$ to measure the effect sizes of reward probability and eye movement direction on neuronal activity, which is not familiar in the field at least for me. Researchers in the field of systems neuroscience often use “area under Receiver Operating Characteristic (ROC) curve” to evaluate to what degree neuron activity

detects a signal in noise. The authors mentioned the reasons why they used ωp^2 in the manuscript. The firstly written reason is that ωp^2 is agnostic to the direction of modulation. That is, this index represents the magnitude of signal but not its direction. For example, some neurons have higher firing rate when reward probability was high, while others have higher firing rate when reward probability was low. This index does not reflect such modulation direction. However, the direction of modulation is very important. Indeed, neurons with different modulation directions sometimes show distinct electrophysiological properties. I strongly recommend the authors to use the well-accepted index, area under ROC curve. Otherwise, skeptic readers don't believe their results.

We appreciate the reviewer's concerns regarding the directionality of responses and have addressed these concerns by adding a supplementary figure showing the fraction of neurons with significant modulation in each direction as a function of time (Fig. S3).

We thank the reviewer for suggesting the use of the area under the curve (AUC). We agree that it is an appropriate effect size measure in many cases; however, we argue that ω_p^2 is more appropriate to our specific purpose. First, AUC does not naturally extend to cases where there is more than one category, such as the eight directions in our study. Secondly, assessing the strength of the interactions is difficult using AUC. In contrast, ω_p^2 allows us to quantify the amount of variability that can be explained by the modulation of one variable over the other. For example, in Fig. 5D, we quantify the extent to which neurons code for prediction error by calculating the ω_p^2 of the reward probability x reward outcome interaction. Thirdly, AUC does not allow us to correct for the variance arising from additional variables in the same epoch. When examining the coding of reward probability during the motion epoch (Fig. 4), the different directional sensitivities of the population (Fig. 3) would affect the calculated reward probability AUC, since they would be treated as noise by the analysis. Fourthly, ω_p^2 also considers dynamics in time, while AUC requires a summary of the data (such as the mean response in a bin). Although the AUC could have been used in some cases, such as the cue epoch, we prefer a single measure that can be used throughout the paper as a whole. When both analyses could be applied, we compared the AUC and ω_p^2 and showed that they are strongly correlated (Fig. RR1, Spearman correlation, all $p < 0.001$).

Fig. RR1: ω_p^2 effect sizes are correlated with the AUC in the cue epoch. **A-B**, Each dot represents a single neuron. In all plots, the horizontal position of the dots represents the ω_p^2 reward probability effect size during the cue epoch. The vertical position of each dot represents the area under the curve of the Receiver-Operating Characteristic analysis decoding the reward probability from the mean firing rate in the cue epoch. **A** shows neurons from the vermis and **B** shows neurons from the basal ganglia. **C**, Bar plots show the averages and standard deviations of the mean for the y-axis. The dashed line represents the identity line.

Minor:

1. The authors show only ω_p^2 for population analyses. I would like to see population (averaged) PSTHs of each type of neurons, which are closer to raw data, because the profile of ω_p^2 data seems to be different from that of conventional electrophysiological data. For example, it has been well known that many neurons in the basal ganglia including the caudate respond to outcome. But, in the present study, the response of caudate neurons to reward outcome, which is shown as ω_p^2 , is tiny. Readers including me may become suspicious about the quality of data.

We included figures in our response that show the average responses of the different populations (Fig. RR2, RR3, and RR4). We note, however, that in the context of the current paper, this analysis has critical limitations. First, some neurons in our sample displayed complex patterns in time^{1,2} (for example, the SNpr neuron shown in Fig. S7). The preferred direction of such cells depended heavily on the specific time window chosen for its calculation, making measures such as tuning curves sensitive to specific parameters. Second, the four populations in our sample were different in their trial-by-trial variances, which means that the modulation of an experimental variable on the average firing rates of neurons in a population might not necessarily correspond to the amount of information that they carry. Finally, the baseline properties of neurons are correlated with rate modulations. For example, neurons with high firing rates tend to have a large rate modulation but also more trial-by-trial variability (see Fig. RR6 and the corresponding response to Reviewer 2). Thus, comparison

based on the population averages might reflect the baseline properties of the neuron rather than the magnitude of the modulation. We showed this is not the case for the ω_p^2 (Fig. S5). Thus we decided to use ω_p^2 , which reflects the signal-to-noise ratio of the neurons.

For these reasons, we were also concerned that the population averages might confuse the reader, so we prefer not to include them. We would, however, gladly include these figures in the paper's supplementary material at the reviewer's request.

Fig. RR2: Population PSTHs in the cue epoch. A-D, Population PSTHs for Purkinje cells (21/90) (A), vermis local neurons (54/134) (B), SNpr neurons (67/231) (C), and caudate neurons (57/232) (D) that significantly differentiated between reward probability conditions. Solid traces are averaged over the reward probability condition in which the firing rate was higher in the cue epoch. Dashed lines show the average over the other condition. The total average firing rate, calculated over both conditions, was subtracted from the PSTH of each neuron.

Fig. RR3: Population PSTHs and tuning curves in the motion epoch. A-D, Population tuning curves of significantly directionally tuned neurons in the pursuit task, aligned to their preferred directions. E-H, Population PSTHs in the pursuit task of the same neurons. A and E show Purkinje cells (34/71), B and F vermis local neurons (51/103), C and G SNpr neurons (117/166), and D and H caudate neurons (62/161). I-L, Population tuning curves of significantly directionally tuned neurons in the saccade task, aligned to their preferred directions. M-P, Population PSTHs in the saccade task of the same neurons. I and M show Purkinje cells (24/65), J and N vermis local neurons (45/87), K and O SNpr neurons (108/150), and L and P caudate neurons (80/169). Red and blue traces correspond to the two reward conditions. In the PSTH plots, solid lines correspond to PSTHs in the preferred direction of the neurons, and dashed lines to the null direction (180° to the preferred direction).

Fig. RR4: Fraction of responsive neurons and population PSTHs in the outcome epoch. **A** and **B**, The fraction of neurons that had a significantly higher firing rate in the no-reward condition (**A**) or the reward condition (**B**) in 50 ms bins (one-tailed rank-sum test). **C-F**, Population PSTHs for Purkinje cells (43/90) (**C**), cerebellar local neurons (68/134) (**D**), SNpr neurons (65/231) (**E**), and caudate neurons (68/232) (**F**) that significantly differentiated between reward outcome conditions. Solid traces are averaged over the reward probability condition in which the firing rate was higher in the cue epoch. Dashed lines show the average over the other condition. The total average firing rate, calculated over both conditions, was subtracted from the PSTH of each neuron.

2. I would like to see the latencies of the reward- and eye movement-related signals in the caudate, SNpr, and the vermis in the cerebellum. The differences in these signal latencies may provide important insights to understand functional segregation between the basal ganglia and cerebellum.

We made an effort to thoroughly investigate the latencies of the different neuronal populations, however, we obtained inconsistent results with minimal effects, which also

varied depending on the specific methodology employed. Consequently, we decided not to include these findings in the paper.

In Fig. RR5, we provide a comprehensive summary of our analyses. We employed two distinct methods to estimate the response latencies of the neurons³. In the first method, we determined the latency of a neuron's response individually for each direction condition. Specifically, we identified the time at which the neuron's activity deviated from the baseline activity by more than 5 standard deviations. Our observations revealed longer latencies in the basal ganglia population than in the vermis on the saccade task, with small yet significant differences on the pursuit task. Conversely, our second method involved defining the latency based on the significance of the difference between conditions. Here, the disparities between populations were minimal and failed to reach statistical significance.

Considering the small magnitude of these differences, possibly due to the inherent noise associated with latency estimation, and their inconsistency with the latencies depicted in Fig. 3, we opted not to include them in the current version of the paper.

Fig. RR5: Response latencies in the motion epoch. **A**, Histograms of latencies of neurons and direction combinations on the saccade task, calculated with the first method. **B**, Latencies, calculated with the first method, as a function of the direction effect size, binned into quantiles (20 bins) by the effect size. **C-D**, Similar analysis as **A** and **B**, for the pursuit task. **E**, Histograms of latencies on the saccade task, calculated with the second method. **F**, Latencies, calculated with the second method, as a function of the direction effect size, binned into quantiles (20 bins) by the effect size. **G-H**, Similar analysis as **E** and **F**, for the pursuit task.

3. The results obtained by the convergence model is described in the discussion section, not in the result section. I feel the description is short and not enough to fully

understand the model and results. I recommend the authors to add new sub-section in the result section to explain details of the model and results.

We removed the convergence model from the manuscript based on feedback from other reviewers.

Reviewer #2 (Remarks to the Author):

In this paper, Larry and colleagues study the representation of reward and movement signals in the cerebellar vermis and the basal ganglia in awake behaving primates. The authors conclude that both these structures have different organization for reward and movement representations. This is an important area of research because 1) it adds more evidence for reward processing in the cerebellum and tries to elucidate mechanisms of reward processing that are currently poorly studied and 2) more importantly, the paper studies 2 brain areas and compare their properties in terms of reward and movement which further sheds light on circuit level properties. The experiments build on previously published task and have similar experimental design and set up. However, I have several conceptual and analysis based concerns that significantly dampen my enthusiasm about the work. This is laid out below. This work needs significant revision to be considered for publication in my opinion.

Major points:

1. My biggest concern about this paper is the disconnect between the question asked and area of the Cerebellum studied. I really doubt if the vermis is the right place to study the questions the authors pose. Although vermis is undoubtedly involved in saccade and pursuit, as the authors noted in this and their previous paper (Lixenberg et al., 2020), the vermis and the flocculus are more tightly connected with the motor system than the hemispheres and thus don't represent reward in their signal as much, simply because of the anatomy. In fact, the hemispheres, for example crus I and II are monosynaptically connected with areas of basal ganglia that process reward (Bostan et al 2010). And Purkinje cells in areas within crus I and II are also shown to process reward (Sendhilnathan et al., 2020 and 2022 - although these are hand related Purkinje cells). In the light of such recent literature, areas within the hemispheres that drive eye movements seem to be a more suitable candidate for the questions the authors ask here. However, if the question were not 'which cerebellar area processes reward in conjunction with the basal ganglia' but rather 'does vermis process reward', then the authors' approach seems more reasonable. Even then, studying a part of the cerebellum (vermis) that is not predominantly involved in reward processing due to its lack of anatomical connections with reward processing areas of the brain including

basal ganglia and concluding that it does not represent reward as much, seems very unsurprising and lacks novelty.

We acknowledge that other parts of the cerebellum and basal ganglia may code for reward to different extents. However, our study aimed to investigate how reward is multiplexed within eye movement regions of the basal ganglia and the cerebellum. To achieve this goal, we recorded from oculomotor areas within both structures. This allowed us to directly compare neurons that participate in producing the same behaviors.

Our initial expectations were indeed in line with the reviewer's comment. We predicted a strong link to movement and weak reward signals in the cerebellar populations in comparison to the basal ganglia. However, this is not what we found. Specifically, we found that reward and movement signals were the smallest in the basal ganglia input, intermediate in the cerebellum, and the largest in the basal ganglia output structures. We believe these results are novel and important regardless of what other areas in each structure might do.

Although strongly linked to eye movement, the vermis has been suggested to be involved in higher-order cognitive and affective functions⁴⁻⁶. Anatomically, some studies have found that the vermis is connected to the limbic system and frontal brain regions, although not in primates^{7,8}, further suggesting a role in cognitive and affective processing. Thus, our findings of reward signals in the vermis and how they interact with movement signals are important in this context as well.

In the revised paper we highlight our approach and explain our decision to compare oculomotor regions across structures. We explain the considerations motivating our choice to focus on the vermis. Additionally, we have included a discussion of other regions of the cerebellum and basal ganglia that may be involved in reward processing, to provide a broader perspective on this topic.

2. Please substitute "cerebellum" with the precise location studied (i.e vermis) throughout the paper. Concluding that 'cerebellum does not represent reward expectation as much' is completely unwarranted since vermis is not representative of the entire cerebellum. This is as if one is recording from V1 and concluding that the 'cortex does not represent faces'. Recording from IT and not V1 will enable researchers to study face representations – similarly recording from an appropriate area of the cerebellum would enable studying reward representations in the cerebellum.

We agree. To prevent over-interpretation of the results we replaced "cerebellum" with "vermis" as the reviewer suggested.

3. Where exactly were the recording made in the vermis and basal ganglia? There's no description or MRI or electrode tract tracing of where the recording was made from the vermis or basal ganglia. This makes the interpretation of results a bit challenging. Furthermore, were neurons from basal ganglia and the cerebellum simultaneously recorded? This wasn't clear anywhere.

We thank the reviewer for this input. In the revised paper we include a supplementary figure (Fig. S1) displaying the MRI and have expanded the methods section explaining how we located our regions of interest. As we specify in the Methods section, we first recorded from the caudate and then from the SNpr, all the while recording from the vermis.

4. I don't see the utility of the convergence model. I am not sure what new information one learns from it. As I understand it, it says as the input signal gets more homogenous, the output signal gets "sharper". But isn't this trivial? More heterogenous input would lead to more averaged output signal and thus, less sharp signal. Furthermore, several physiological aspects of this model are not clear. First, the blue and red neurons are hypothesized to be neurons that respond to .75 or .25 reward probability. However, did authors really find these neurons in their ephys explorations? The only example comes from fig 1 but those neurons don't have impressive differences except for 1F - especially since the spike density functions lack error bar, it's hard to tell if the differences are actually significant or not. Second, the model by Herzfeld et al 2015 suggests that only P-cells with similar properties (CS direction) converge to the same nucleus neuron and this best explains saccade error direction coding in cerebellum. The model in the current manuscript seems to be a generalized version of the Herzfeld model but I am not convinced how the heterogeneous aspect will work out due to "less sharp" signals. Third, and related to the above point, what is the relationship between CS direction and the blue and red neurons here? Apart from this, what is the motivation for the model? Is it to explain the ephys findings or to make any prediction? I am not sure if it does either. In short, I don't think I learn much from the model and it raised more questions than provide answers to me.

We agree, and based on this feedback and other comments, we have removed the model from the paper and replaced it with a more extensive discussion that we hope will provide a better understanding of the results and their implications. We have also added error bars to the PSTHs in Fig. 1.

Minor points:

1. In line 99-100 the authors mention they identify neurons that are reward selective. How was this done? By eyeballing or using any statistical test? Please describe this method.

We used ω_p^2 to quantify the responses of all neurons in our sample, without an initial selection based on reward selectivity. We modified this specific line to avoid confusion, and also state it clearly in the text (line 89).

2. Does the effect size depend on both 1) difference in the value of the variable for a given condition (for example .75 reward) relative to noise and 2) the trial-to-trial difference in the activity within a condition? In that case, if the effect size is low, it could be because of either or both of the above reasons. How do the authors disambiguate it?

ω_p^2 is an unbiased estimator of the ratio between condition variance and the condition variance plus the within-condition (unexplained) variance, and can be viewed as an estimator of the signal-to-noise ratio (SNR). Thus, ω_p^2 will increase if the average difference between conditions increases, or if the trial-by-trial variability increases. Disentangling the two possibilities is a fundamental problem in SNR measures or measures that rely on the SNR such as the area under the curve or decoders, since an increase in trial-by-trial variability will also map to larger sampled differences between condition averages, due to larger sampling errors.

To demonstrate this point using our data, we generated a plot (Fig. RR6) that compares the sum of squares of the direction variable between conditions to the sum of squares within conditions during the motion epoch. Notably, these two measures exhibited a strong correlation (Spearman correlation: $r=0.88$, $p<0.0001$, $n=678$). Thus, in contrast to ω_p^2 (Fig. S5), the signal and noise characteristics of the neurons in our study are influenced by the firing rates. Thus, comparing them between populations is essentially a comparison of the mean firing rates of neurons.

Fig. RR6: Correlation of the sum of squares within and between conditions and the firing rate. Each dot represents the sum of squares between direction conditions (horizontal) and the sum of squares within conditions (vertical) of a single neuron in the motion epoch on a

log-log scale. The color represents that neuron's firing rate in the same epoch on a logarithmic scale.

3. Related to the point above, since SNR for CS is low, and since they only fire in about 30-40% trials, the trial-to-trial variability is also high and therefore I highly doubt if the effect size analysis is the right way to analyse the CS data. Furthermore what exactly is the conclusion of the CS analysis?

Fig. 6 was added to provide a more comprehensive depiction of the signals within the vermis. Although the firing rate of complex spikes is low, a low rate will not necessarily result in a low signal-to-noise ratio (Fig. S5). If the firing rate is low but the response is consistent, it can still result in a high effect size. In this analysis, we demonstrate that the complex spike activity of individual Purkinje cells conveys little information about reward and direction on a trial-by-trial basis compared to their simple spike activity. We also included a more conventional analysis of complex spikes that average across the population (Fig. S9).

4. What are “local neurons”? Golgi, interneurons etc? Please define them properly since different classes of neurons in the cerebellar microcircuit have different spike waveforms, inter spike interval distributions and different functions.

We agree with the reviewer that differentiating between local cerebellar populations is important. In an attempt to classify the different neurons in our sample into the various vermis subgroups, we analyzed the average firing rate and coefficient of variation (CV) during fixation (the cue epoch)⁹, as well as the width of the action potential waveform (through-to-peak)¹⁰. The scores of individual local neurons on the first two principal components (PCs), which explained more than 75% of the variance, are shown in Fig. RR7. However, we were unable to identify any clear clusters when examining the 2D or the 3D data (not shown). Therefore, we refrain from speculating on the exact cell type of the local neurons and instead focus on features that we found to be related to their encoding properties. The results of this analysis are presented in the revised paper (Fig. 6). In brief, we found that neurons that were strongly directionally tuned had low firing rates and narrow waveforms.

Fig. RR7: Principal component analysis of local vermis neurons. A and B, The horizontal location of each dot represents a single local vermis neuron's score on the first PC. In **A**, the vertical location represents the score of the neurons on the second PC. In **B**, the vertical location represents the direction effect size. The r value represents the Spearman correlation ($p=0.06$, $n=128$).

5. line 288 typo – reanalysis

We corrected the typo.

6. Lines 206-207 although the neurons are tuned to saccade direction as well as pursuit directions, is the coding direction the same for saccade and pursuit? This is not clear.

The effect size analysis does not assess the similarity or differences in the ways direction is encoded between the two tasks. To make this comparison, one could calculate the differences between the preferred directions, defined as the center of mass of the neuron's tuning curve, on the two tasks. Fig. RR8 demonstrates that preferred directions tended to be similar in all populations except Purkinje cells. A more comprehensive comparison between tasks, however, would require an analysis of the temporal dynamics of the neurons, since the dynamics of the response may be crucial in encoding direction¹¹. In a recent preprint, we showed that neurons often have complex temporal responses², such as the SNpr neuron in Fig. S7, which makes the preferred direction highly dependent on the specific time window used to calculate it. Thus, the results in Fig. RR8 were very sensitive to the time window of the analysis, and the tuning direction was inconsistent throughout the trial. We believe that our dataset can shed light on the encoding of saccades and pursuit, but that a rigorous comparison of the neural dynamics between movements with vastly different temporal profiles requires further exploration. We think this is beyond the scope of this manuscript, and we plan to study this topic in more detail in future work.

Fig. RR8: Differences between preferred directions in the pursuit and saccade tasks, Differences between the preferred directions (PD) in the saccade and pursuit tasks, for neurons recorded in both, which significantly differentiated between direction conditions on

at least one task (Kruskal-Wallis test, Purkinje cells (28/46), local vermis neurons (38/56), basal ganglia SNpr neurons (74/85), and caudate neurons (56/98)).

7. Are the plots in Fig S6 only for correct trials? Since trial outcome and reward outcome can be disassociated in this task, I wonder how the neural activity would look for these two different parameters.

All analyses in this paper focus on trials in which the monkeys successfully completed the trial. In a small fraction of the trials, which were excluded from our analysis, the monkeys failed to complete the trial. We excluded these trials since failures can arise from a range of factors such as early cessation of fixation, blinking, or not accurately tracking the target, making their analysis beyond the scope of our paper.

8. The discussion feels insufficient. Greater insights can be drawn from the results. Currently the discussion is heavily dominated by the model which doesn't seem that useful in my opinion (see major point 4). The authors record from two brain areas here – what can be said not just about these areas individually but the relationship between them at the level of a circuit?

We removed the model and extended the discussion.

Reviewer #3 (Remarks to the Author):

This study in behaving macaques aimed to compare activity of neurons in the basal ganglia and cerebellum of the same monkeys using the same tasks, thus providing consistency across the neural samples that helps to interpret the relative functions of the areas. The tasks varied probability of receiving reward for correct responses using an instructional cue, as well as direction of saccadic and smooth pursuit eye movements. The study is presented in a framework that purports to study the inputs and outputs of both the basal ganglia and cerebellum, so as to facilitate understanding of the signal transformations within each subcortical network. The key hypotheses are that reward signals are more prevalent at the inputs while movement direction signals are more prevalent at the outputs, and that the cerebellum has more of a movement role than the basal ganglia. Large samples of neurons were recorded in the caudate and substantia nigra pars reticulata (input and output of basal ganglia, respectively) and non-Purkinje cells and Purkinje cells of the cerebellar cortex (input and output of cerebellum, respectively). In general, the hypotheses were refuted in ways that are potentially very interesting and highlight newly appreciated differences between basal ganglia and cerebellar processing.

Overall, the manuscript provides a wealth of data that will be interesting to experts. The data are collected and analyzed rigorously, for the most part. The framework as described above

is interesting, and if tested specifically to compare analogous processing zones of the basal ganglia and cerebellum, could yield compelling results. The choice of basal ganglia recording regions (caudate and SNr) conform to the framework well, but the choice of cerebellar cortical neurons do not. The report presents a worthy dataset but I have concerns that the weakness in matching regions of study to the conceptual input-output framework may limit the general impact of the work.

Big picture comments:

1. The main cerebellar neurons studied were Purkinje cells. These are neither at the input nor the output of the cerebellum. The paper positions them at the output. This is true for cerebellar cortex, but not for cerebellum as a whole. Purkinje cells innervate neurons of the deep cerebellar nuclei (and some other targets), that are considered to provide cerebellar output to the rest of the brain.

Moreover, the paper positions the non-Purkinje cell neurons in cerebellar cortex as comprising an input stage. First, they are not the input stage; it is generally accepted that granule cells are. But they are not even necessarily open-loop inputs to the Purkinje cells. Collaterals from Purkinje cell axons are known to feedback to and modulate the cerebellar cortex interneurons.

A few references:

- Crook, J.D., Hendrickson, A., Erickson, A., Possin, D. and Robinson, F.R., 2007. Purkinje cell axon collaterals terminate on Cat-301+ neurons in Macaca monkey cerebellum. *Neuroscience*, 149(4), pp.834-844.
- Hirono, M., Saitow, F., Kudo, M., Suzuki, H., Yanagawa, Y., Yamada, M., Nagao, S., Konishi, S. and Obata, K., 2012. Cerebellar globular cells receive monoaminergic excitation and monosynaptic inhibition from Purkinje cells. *PLoS one*, 7(1), p.e29663.
- Guo, C., Rudolph, S., Neuwirth, M. E., & Regehr, W. G. (2021). Purkinje cell outputs selectively inhibit a subset of unipolar brush cells in the input layer of the cerebellar cortex. *Elife*, 10, e68802.
- Witter, L., Rudolph, S., Pressler, R. T., Lahlaf, S. I., & Regehr, W. G. (2016). Purkinje cell collaterals enable output signals from the cerebellar cortex to feedback to Purkinje cells and interneurons. *Neuron*, 91(2), 312-319.

In other words, the cerebellar cortical interneurons and Purkinje cells do not necessarily mediate a serial flow of information, but may form a closed loop microcircuit. This is not addressed.

I recognize that achieving the ideal match of input and output stages between basal ganglia and cerebellum would be exceedingly difficult in monkeys, primarily because the

granule cells of the cerebellum (the input stage there) are technically challenging to study. But it does not seem infeasible to match the output stages better, i.e. SNr and a deep cerebellar nucleus. Put simply, the framework did not seem to drive the selection of recording sites; it appears more that an impressive data set, across multiple areas, was collected and then a framework was constructed to justify direct comparisons across the areas. But due to the mismatch of the cerebellar sites to the input-output concept, that framework is not really appropriate.

I do emphasize, however, that this is a beautiful data set. It will be of great interest to basal ganglia and cerebellar researchers. It may be more straightforward to present the data more plainly, with a more scholarly discussion of what is learned from the comparisons that can be made and the limitations of the comparisons.

We appreciate the reviewer's acknowledgment of the significance of our dataset. We agree that the cerebellar cortex is not accurately described as a feedforward network. For that reason, in the previous version of the manuscript we referred to the non-Purkinje cerebellar cortex population as "local" neurons and not "input" neurons. To avoid over-interpretation of our claims regarding the flow of information within the cerebellar cortex, we removed the model and replaced it with a more extensive discussion.

In the revised Discussion section, we explore various potential explanations for the apparent absence of signal sharpening in the cerebellum, one of which involves the influence of Purkinje cell collaterals on the cerebellar cortex. We have also expanded the discussion to include other components of the basal ganglia and cerebellum, such as the deep cerebellar nuclei. The revised paper now places less emphasis on the comparison between input/local and output stages.

2. The results in basal ganglia not only refute the hypothesis that neural signals transition from more reward-related to more movement-related from caudate to SNr (which is interesting), but also throw a paradoxical monkey wrench into the framework of caudate as input and SNr as output. The caudate signals are not only weaker, they are later than the SNr signals. As pointed out by the authors, this could mean that the SNr is receiving substantial inputs of non-caudate origin. That is also very interesting, but if the framework breaks down for basal ganglia, and did not seem to motivate the selection of recording sites in the cerebellum, it is unclear how the use of the framework promotes clear communication of the results.

We thank the reviewer for these comments and have made a major revision of our paper accordingly. We have reduced the emphasis on the transition of information within structures and instead place greater focus on the respective roles of the cerebellum and basal ganglia in

processing reward-related and movement-related signals. We have expanded the discussion to include alternative sources of information that contribute to the SNpr.

Major data comments:

1. It is interesting that no region appears to encode reward prediction error (RPE) strongly, although, as the authors note, the study does not appear to be specifically designed for this, as this epoch is confounded with sipping. Furthermore, because the RPE would occur at reward delivery, the animal no longer needs to fixate, so it is unclear what the eyes are doing in this period (perhaps staring at the juice tube). With the current study design, it is unclear when the animal should experience a negative RPE in the case of reward omission (and may end up smearing any signal across time) because the animal must infer the omission through a timing task – comparing a noisy estimate of how much time has passed since task completion to a noisy expectation of when the reward should arrive by if delivered.

I don't think one can speak to coding of reward outcome with this task design (Fig 4). The eye could be doing anything in this phase, so a saccade from the target to the juice tube could account for directional encoding at this epoch (Fig 4B) and differences between what the eyes do for reward delivery/omission could account for the reward outcome effect (Fig 4C). If the authors can account for condition-dependent eye movements/position during this epoch, that could address this problem.

As pointed out by the reviewer, our study did not find evidence for a reward prediction signal in any of the neuronal populations analyzed. However, we did identify a signal that correlated with the reward outcome (i.e., whether a reward was delivered or not). We rigorously analyzed the outcome epoch. Several observations indicate that the coding of reward prediction error is not spread out in time. First, the end of each trial is clearly marked by the disappearance of the colored target, and the reward was delivered (or omitted) immediately thereafter. Thus, the monkeys could infer whether or not they received a reward right after the trial ended. Secondly, we observed rapid changes in the monkeys' licking patterns between rewarded and unrewarded trials (Fig. RR9), suggesting a short latency for reward outcome detection. Thirdly, we did not find differences in neural activity between trials with different reward probabilities ($P=0.25$ vs. $P=0.75$), despite the clear auditory signal of the reward pump operating in these trials. We addressed this point in our revised paper (line 236).

Fig. RR8: Licking behavior in the outcome epoch. A-B, Fraction of trials with licks during the outcome epoch for the P=0.25 condition (A and B) and the P=0.75 condition (C and D). Solid lines correspond to the reward conditions and the dashed line to the no-reward conditions. A and C show the behavior of monkey A, and B and D show the behavior of monkey G.

We concur with the reviewer's assessment that the interpretation of the outcome signal is complicated due to the occurrence of uninstructed behaviors at the end of each trial. To address this concern, we conducted a thorough analysis of two of these behaviors; namely, blinks and saccades, which we recorded. This analysis yielded slight differences in eye movement and blinking across task conditions. However, as we show below, these effects were either inconsistent with the coding of the outcome or too small to underlie the coding of the outcome. Thus, the saccades and blinks do not underlie the coding of the reward outcome. We refer to this in the revised paper.

We observed that monkeys tended to blink at the end of trials (Fig. RR10A and B). They also saccaded back towards the center of the screen, where a new fixation point would appear to initialize the upcoming trials (Fig. RR10C and D). We found that blinks were more frequent when the reward was omitted than when it was delivered. In contrast, early saccades at the end of trials were more frequent when a reward was delivered than when it was omitted. Several lines of evidence indicate that blinks and saccades did not underlie the coding of the reward outcome signal. First, we did not observe an outcome x direction interaction in this epoch (Fig. RR11E). This suggests that if the outcome effect size was driven by the coding of saccades, this coding was not modulated by the direction of the saccades. We find this unlikely, as many neurons in all populations were directionally tuned, as demonstrated in Fig. 3.

Secondly, we repeated our analysis of the reward outcome signals while excluding trials with blinks in the first 500 ms, and equating the number of saccades in the reward/no-reward conditions by discarding trials with and without early saccades until the numbers were equal. Fig. RR11F shows that there are outcome effects even under these conditions (Bootstrap t-test: $p < 0.001$ for all populations). These findings suggest that the observed outcome signals were not driven by the monkeys' blinks or saccades.

Fig. RR10: Blinks during the outcome epoch. **A** and **B**, Blink rates for monkeys A (**A**) and G (**B**) aligned to the end of the trial for the different reward probability and reward outcome conditions. **C** and **D**, Saccade rates for monkeys A (**C**) and G (**D**) aligned to the end of the trial for the different reward probability and reward outcome conditions. **E**, Direction x reward outcome interaction effect sizes in the outcome epoch. **F**, Reward outcome effect sizes in the outcome epoch in trials without blinks and after the number of early saccades was equated.

Thirdly, we could not explain the coding of the reward outcome by the amplitudes of the saccades. Specifically, we observed that Monkey A tended to make smaller amplitude saccades in the P=0.25 no-reward condition compared to the P=0.25 reward condition and both P=0.75 conditions (Fig. RR11A). On the other hand, Monkey G tended to make smaller amplitude saccades in the P=0.25 reward condition and the P=0.75 no-reward condition

compared to the other two conditions (Fig. RR11B). If the neural activity had followed the patterns of saccade amplitudes in either monkey, we would have expected an interaction between reward probability and outcome. However, the interaction effect size was small, as illustrated in Fig. 5D. To focus on the contribution of these patterns, we calculated ω_p^2 for the corresponding amplitude patterns of each monkey. For Monkey A, we calculated the effect size of the contrast P=0.25 no-reward versus P=0.75 no-reward. For Monkey G, we calculated the effect size of the contrast P=0.25 no-reward and P=0.75 reward versus P=0.75 no-reward and P=0.25 reward. We observed a small effect size in both cases (Fig. RR11C and D). Overall, these analyses suggest that the coding of the reward outcome signals we observed was not driven by the monkeys' saccades at the end of trials.

Fig. RR11: Saccades during the outcome epoch. **A** and **B**, Histograms of the amplitudes of the first saccades after the end of the trials for monkeys A (**A**) and G (**B**) for the different reward probability and reward outcome conditions. **C**, P=0.25 no-reward versus P=0.75 no-reward contrast effect size for neurons recorded from monkey A. **D**, P=0.25 no-reward and P=0.75 reward versus P=0.75 no-reward and P=0.25 reward contrast effect size for neurons recorded from monkey G.

2. In the saccade task the target always appeared at 10 degrees. The authors don't mention the receptive/movement field eccentricity or size for the regions and neurons they recorded from. With the saccade targets always at 10 degrees, the measure of directional coding would seem to insufficiently characterize neurons that may have relatively punctate, diagonally aligned, or smaller eccentricity regions of sensitivity. This suggests that differences in response field locations or sizes across the recording sites may account for some of the differences between cerebellum/caudate vs. SNr directional encoding.

We agree with the reviewer that differences between the size and eccentricities of the visual receptive and movement fields between the populations are important factors to consider. Although we did not measure the receptive or movement field of neurons, we opted to include two different eye movement behaviors to demonstrate that our results are not specific to the parameters chosen in our experiment. In the pursuit task, the target remains close to the eye position throughout the trial, while in the saccade task, the target appears abruptly in the periphery of the visual field. We found similar results on both tasks, indicating that the average coding of direction was similar between these two different eye movements (Fig. 3A and B). Additionally, in all populations, the direction effect sizes were correlated between tasks (Fig. 3C and D). Thus, it appears that the extent to which individual neurons code for direction, relative to other neurons, is independent of the specifics of the eye movement.

We discuss this important point in the revised Discussion section (line 310).

3. While omega squared seems to be a reasonable choice for quantifying reward signals, in the case of directional encoding, the authors should provide some explanation to justify its use. There are many ways neurons can encode movement beyond directional sensitivities. Measures of timing and vigor, for example, may contribute additional useful information for comparing across areas.

In this study, we examined the coding of neurons on a trial-by-trial basis. This required multiple repetitions of the same condition and thus limited the number of variables that we could manipulate given the constraints of recording from the same neurons. We decided to manipulate two variables in the motion epoch of the task: reward probability and direction, and compare neurons across two different eye movements (Fig. 3 and Fig. 4).

While we acknowledge that studying additional motor parameters such as pursuit velocity and saccade amplitude would be an important future direction, we did not manipulate these variables in this study. As a result, we would have to rely on smaller differences that occurred spontaneously between conditions (Fig. S2), which would have made the results very difficult to interpret.

We used ω_p^2 as an estimator of the extent of directional sensitivities for several reasons. First, it is an unbiased estimator, allowing us to conclude that a population does not respond to direction when it is zero on average. Second, it enabled us to separate the coding of reward probability from trial-by-trial noise, thus facilitating comparisons between populations that respond to probability to varying degrees (Fig. 4). Third, it accounted for the dynamics in time, making it possible to analyze complex patterns over time, such as the SNpr neuron in Fig. S7².

We also analyzed the differences in latency between populations. We included an examination of response latencies in our response to Reviewer 1's first minor comment (Fig RR5).

Minor comments:

1. The manuscript is missing some important behavioral measures. How did performance (in terms of % correct) in the 25% reward probability condition compare to the 75% reward probability condition? Did animals have any directional biases in terms of performance and reaction time? Different reaction times across different target/target motion directions could create artificial differences in how neurons respond to each direction. This is primarily a problem because Fig 3A/B are aligned to target motion, not saccade/pursuit onset.

We include a more detailed analysis of the pursuit and saccade behaviors in the revised paper (Fig. S2).

We agree with the reviewer that anisotropies are important to take into account. To verify that differences in saccade reaction time or pursuit latency cannot account for the differences in direction effect sizes between the populations, we calculated the direction effect sizes on data aligned to the saccade or pursuit initiation (Fig. RR12). We found similar results to the ones shown in Fig 3 (Fig. RR12 and B). In the saccade task, there was an instantaneous increase in the average direction effect size of the local vermis population. However, the effect sizes were similar between alignments (Fig. RR12C-D).

Fig. RR12: Coding of eye movement direction during pursuit and saccades, aligned to eye movement onset. **A** and **B**, Direction effect size using the target motion epoch aligned to the pursuit set in the pursuit task (**A**) and the saccade onset in the saccade task (**B**). **D-F**, The horizontal location of each dot represents a single neuron's direction effect calculated with the data aligned to motion onset. In **C** and **D** the vertical location shows the effect size aligned to the pursuit, and in **E** and **F** to the saccade. **C** and **E** show the vermis populations and **D** and **F** the basal ganglia. The r values represent the Spearman correlation ($p < 0.0001$ for all populations).

2. Fig 1C task details are lacking. Did the monkey select the stimulus via a saccade? Where were the stimuli placed? A brief description would be helpful here.

In the revised paper we added additional details in the Results section to make the description of the task clearer. We also refer the readers to the Methods section for a more detailed description of the task.

3. The authors find significant differences between behavior in the two reward conditions, but never discuss whether or not this is important in the interpretation of their results. The differences are not an issue in Fig 1A which is before movement, but could impact the

interpretation of Fig 1B, since these small differences in movement metrics could lead to spillover from movement coding into reward coding.

We agree that differences in eye movement could potentially confound the effects of reward during the motion epoch (Fig. 4). We found consistent but small differences in pursuit and saccade behaviors between the reward probability conditions (Fig. S2). Below we show that since these differences were slight, they are unlikely to underpin the coding of reward probability during target motion. We highlight this point in the revised paper (line 193).

In our analysis, we found that the effects of reward probability during this epoch were very small in the vermis and caudate populations. Thus, the impact of the behavioral differences due to reward on the activity in this epoch was negligible. However, in the SNpr population, we observed an increase in the reward probability effect size following target motion. Since this was the only population that showed this increase, we focused on ruling out the confounding effects of eye movement in this population.

In the pursuit task, we found that eye movements were on average slightly faster in the $P=0.75$ condition than in the $P=0.25$ condition. Consequently, there was also a smaller average number of corrective saccades in the $P=0.75$ condition. To verify that there was no spillover from velocity coding, or the coding of corrective saccades, into the reward effect size, we replaced the reward probability variable from our model of the cue epoch with one of two variables. The first was a velocity variable that indicated whether the pursuit velocity in the trial in the 400 to 800 ms after the target motion was faster or slower than the median velocity in the session. We chose this time window since it is the time window in which we observed the largest reward probability effect sizes in the SNpr. The second was a variable that indicated whether the number of corrective saccades was larger or smaller than the median number of corrective saccades in a session. The velocity and saccade effect sizes were smaller than the reward effect sizes (Fig. RR13) indicating that pursuit velocity and corrective saccade number explained less of the variance in SNpr neurons than the reward condition.

In the saccade task, we found that the average saccade latency was slightly longer in the $P=0.75$ condition. The effect of reward probability on other behavioral metrics such as saccade duration, velocity, and end location, and the number of corrective saccades during pursuit was very slight and inconsistent across monkeys. To verify that saccade latency could not explain our results, we aligned the neural activity to the saccade onset and replicated our results (Fig. RR12).

Fig. RR13: Pursuit velocity and the number of corrective saccades coding in the SNpr. **A** and **B**, The horizontal location of each dot represents a single SNpr neuron's reward probability effect size in the motion epoch. In **A**, the vertical location represents the eye velocity effect size during the same epoch (Permutation Welch's t-test: $p < 0.001$). In **B**, the vertical location represents the number of corrective saccades effect size (Permutation Welch's t-test: $p < 0.01$).

4. Line 146: pSNpr-purkinle -> purkinje

We corrected the typo.

5. Lines 224-225: What is this saccade latency relative to – reward, or disappearance of the target?

The reward is delivered when the target disappears and the trial ends. We clarify this in the revised paper (line 211).

6. Fig 1: No error bands, so it's difficult to tell what is noise. Black diamonds are in D, not F (legend line 11).

We added SEM bars and corrected the reference.

7. Fig 3: It would be helpful to show some measure of correlation for plots like C, otherwise I don't know what to make of these plots.

We added dashed lines over the identity line in Fig. 4B (previous Fig. 3C) to highlight the larger direction effect size compared to the reward effect size for most neurons.

8. The model may be better placed in the Results, not the Discussion. It seems to be more than just a “Discussion figure” that provides a take-home message; it involved careful

choice of parameters, selection of model approach, etc, that feels out of place as a concluding point.

Based on the feedback received from the reviewers, we have decided to remove the convergence model from the manuscript.

1. Churchland, M. M. *et al.* Neural population dynamics during reaching. (2012) doi:10.1038/nature11129.
2. Zur, G., Larry, N. & Joshua, M. High-Dimensional Encoding of Movement by Single Neurons in Basal Ganglia Output. *bioRxiv* 2023.05.17.541090 (2023) doi:10.1101/2023.05.17.541090.
3. Levakova, M., Tamborrino, M., Ditlevsen, S. & Lansky, P. A review of the methods for neuronal response latency estimation. *Biosystems* **136**, 23–34 (2015).
4. Schmahmann, J. D. An Emerging Concept: The Cerebellar Contribution to Higher Function. *Arch Neurol* **48**, 1178–1187 (1991).
5. Schmahmann, J. D. The role of the cerebellum in affect and psychosis. *J Neurolinguistics* **13**, 189–214 (2000).
6. Guell, X., Schmahmann, J. D., Gabrieli, J. D. E. & Ghosh, S. S. Functional gradients of the cerebellum. *Elife* **7**, (2018).
7. Harper, J. W. & Heath, R. G. Anatomic connections of the fastigial nucleus to the rostral forebrain in the cat. *Exp Neurol* **39**, 285–292 (1973).
8. Snider, R. S., Maiti, A. & Snider, S. R. Cerebellar pathways to ventral midbrain and nigra. *Exp Neurol* **53**, 714–728 (1976).
9. van Dijck, G., Hulle, V., Heiney, M. M., Blazquez, S. A. & Meng, P. M. Probabilistic Identification of Cerebellar Cortical Neurones across Species. *PLoS One* **8**, 57669 (2013).
10. Johnston, K., DeSouza, J. F. X. & Everling, S. Monkey Prefrontal Cortical Pyramidal and Putative Interneurons Exhibit Differential Patterns of Activity Between Prosaccade and Antisaccade Tasks. *Journal of Neuroscience* **29**, 5516–5524 (2009).
11. Churchland, M. M. *et al.* Neural population dynamics during reaching. *Nature* **487**, 51–56 (2012).

REVIEWER COMMENTS

Reviewer #1 (Remarks to the Author):

In response to my first comment on sampling bias, the authors explained as follows.

“Specifically, after identifying the oculomotor areas, we did not limit ourselves to eye movement neurons. Therefore, the movement and reward signals reported in this manuscript reflect the distribution of the signal within the oculomotor areas.”

This is nice. So, the authors need to show the proportion of the neurons representing pure reward signal, that of the neurons representing pure eye-movement signal, and that of the neurons representing both signals. This information corroborates the conclusion of the present study (i.e., Reward expectation and movement signals were the most pronounced in the output structure of the basal ganglia, intermediate in the cerebellum, and the smallest in the input structure of the basal ganglia.), and helps readers to compare the data of the present study with literature.

The authors removed the description of how to sample neurons in the revised manuscript. But this information is necessary. The authors need to add an accurate description.

The authors properly responded to the other comments.

Reviewer #2 (Remarks to the Author):

I highly appreciate the new analyses, changes and supplementary analyses authors performed to answer my and other reviewers' questions. The manuscript has significantly improved. I have two other comments before I recommend publication.

1. Fig S9B is unclear. Could the plots be made transparent so that we can see the overlays clearly? This figure is especially important to make inferences about reward related CS activity and further supports my original argument that size effect might not be the best way to analyze/visualize Complex spikes.

Related to this, I would rephrase the sentence in line 268. Complex spikes have been to primarily shown to be a teaching signal promoting plasticity only during motor learning. However during reward processing, evidence for such a type of teaching signal has not been shown yet. For example, the authors' own previous work - Larry et al., eLife, 2019 apart from Sendhilnathan et al., Nat Comms 2021, Ohmae et al., 2015 etc. This could also be a short discussion point as currently the CS analyses seem 'unfinished'.

Also, there is a wrong figure reference in line 249.

2. Apart from this, I am still not highly convinced with the response to my original minor comment 2. I understand that the two processes I listed are highly correlated and tricky to disambiguate. I wonder if the advantages of using a new metric like this outweighs its disadvantages such as this issue. Currently there are spike density functions and rasters of examples neurons in Fig 1 which I appreciate. However as a naive reader, I would like to see at least one figure with population plots computed in the traditional way so that it gives me clarity on the 'raw data'.

Reviewer #3 (Remarks to the Author):

The manuscript is substantially improved.

Both of my Big Picture Comments have been addressed. However, in doing so, the authors could not make a strong case that they are comparing input vs output of the cerebellum like they are for the basal ganglia, and thus they changed their overall framework and dropped their model. On the plus side, this means the manuscript is now more accurate in its description of the dataset. But the changes, unfortunately, made the report less intriguing. I still think the dataset is important. It will be of interest to experts in reward and movement. I recognize and emphasize the significance of collecting these physiological results across subcortical areas in the same monkeys in the same tasks. The outcome of the revision, however, was to water down the main takeaway points on how signal flow differs between the basal ganglia and cerebellum. I find the conclusions of the paper moderately compelling but not particularly definitive.

The authors satisfactorily addressed most of my specific concerns about the data and figures, except for one. Regarding my Major Data Comment #2, the authors argued that the target appears in two positions – one foveal and the other peripheral – because of the use of a saccade and pursuit task and that coding between the two tasks is similar. However, the similarity of coding between the two tasks is not as much

as a concern for me as the differences in tuning curves between regions. With the authors' approach, it appears that a neuron with a very sharp tuning curve (on for one direction, off for all others) would show weaker directional coding than a neuron with broader tuning (peak at one direction with graded response across all directions), but it's not clear which case, if any, is coding direction most strongly. Thus, one of the authors' major claims, that the coding of eye movement direction is more pronounced in the SNpr than in the vermis and the caudate, could be a misinterpretation caused by broader tuning curves in SNpr compared to other regions. The authors should address this more carefully in the manuscript.

Reviewer #1:

1. In response to my first comment on sampling bias, the authors explained as follows.

“Specifically, after identifying the oculomotor areas, we did not limit ourselves to eye movement neurons. Therefore, the movement and reward signals reported in this manuscript reflect the distribution of the signal within the oculomotor areas.”

This is nice. So, the authors need to show the proportion of the neurons representing pure reward signal, that of the neurons representing pure eye-movement signal, and that of the neurons representing both signals. This information corroborates the conclusion of the present study (i.e., Reward expectation and movement signals were the most pronounced in the output structure of the basal ganglia, intermediate in the cerebellum, and the smallest in the input structure of the basal ganglia.), and helps readers to compare the data of the present study with literature.

We added a table with this information to the revised paper (Table 1).

2. The authors removed the description of how to sample neurons in the revised manuscript. But this information is necessary. The authors need to add an accurate description.

We compared the two versions of the manuscript and did not find any instances where information on the sampling procedure had been deleted . We are more than willing to reinstate any inadvertently omitted details or incorporate additional information that would enhance the clarity of how the data were collected.

3. The authors properly responded to the other comments.

We thank the reviewer for the input.

Reviewer #2:

1. I highly appreciate the new analyses, changes and supplementary analyses authors performed to answer my and other reviewers' questions. The manuscript has significantly improved. I have two other comments before I recommend publication.

We thank the reviewer for the input.

2. Fig S9B is unclear. Could the plots be made transparent so that we can see the overlays clearly? This figure is especially important to make inferences about reward related CS activity and further supports my original argument that size effect might not be the best way to analyze/visualize Complex spikes.

We modified the figure.

3. Related to this, I would rephrase the sentence in line 268. Complex spikes have been to primarily shown to be a teaching signal promoting plasticity only during motor learning. However during reward processing, evidence for such a type of teaching signal has not been shown yet. For example, the authors' own previous work - Larry et al., eLife, 2019 apart from Sendhilnathan et al., Nat Comms 2021, Ohmae et al., 2015 etc. This could also be a short discussion point as currently the CS analyses seem 'unfinished'.

We agree with the reviewer that further analysis of the complex spike signal analysis could reveal interesting insights. However, as the reviewer suggests, this would require a different approach. Therefore, we believe that this is beyond our current paper's scope, and plan to explore this aspect in depth separately. We remove the sentence since our task does not directly probe learning.

4. Also, there is a wrong figure reference in line 249.

We corrected the reference.

5. Apart from this, I am still not highly convinced with the response to my original minor comment 2. I understand that the two processes I listed are highly correlated and tricky to disambiguate. I wonder if the advantages of using a new metric like this outweighs its disadvantages such as this issue. Currently there are spike density functions and rasters of examples neurons in Fig 1 which I appreciate. However as a naive reader, I would like to see at least one figure with population plots computed in the traditional way so that it gives me clarity on the 'raw data'.

We added supplementary figures showing the averaged population responses (additional plots in Fig. S3, new figures S8 and S10).

Reviewer #3:

1. The manuscript is substantially improved. Both of my Big Picture Comments have been addressed. However, in doing so, the authors could not make a strong case that they are comparing input vs output of the cerebellum like they are for the basal ganglia, and thus they changed their overall framework and dropped their model. On the plus side, this means the manuscript is now more accurate in its description of the dataset. But the changes, unfortunately, made the report less intriguing. I still think the dataset is important. It will be of interest to experts in reward and movement. I recognize and emphasize the significance of collecting these physiological results across subcortical areas in the same monkeys in the same tasks. The outcome of the revision, however, was to water down the main takeaway points on how signal flow differs between the basal ganglia and cerebellum. I find the conclusions of the paper moderately compelling but not particularly definitive.

We thank the reviewer for acknowledging the importance and uniqueness of these data. Recent findings of reward signals in the cerebellum have relaunched the debate on the

distinction between the roles of the basal ganglia and cerebellum¹⁻⁴. We view this discussion as critical to our understanding of the motor system, and consider that this manuscript contributes to addressing it. We agree that additional experimental and theoretical research will have to be conducted to revise our model of the subcortical motor system. However, the current manuscript provides the first comprehensive and direct comparison of reward and motor signals across these structures. We believe that this comparison will be of considerable interest to the motor system field.

2. The authors satisfactorily addressed most of my specific concerns about the data and figures, except for one. Regarding my Major Data Comment #2, the authors argued that the target appears in two positions – one foveal and the other peripheral – because of the use of a saccade and pursuit task and that coding between the two tasks is similar. However, the similarity of coding between the two tasks is not as much a concern for me as the differences in tuning curves between regions. With the authors' approach, it appears that a neuron with a very sharp tuning curve (on for one direction, off for all others) would show weaker directional coding than a neuron with broader tuning (peak at one direction with graded response across all directions), but it's not clear which case, if any, is coding direction most strongly. Thus, one of the authors' major claims, that the coding of eye movement direction is more pronounced in the SNpr than in the vermis and the caudate, could be a misinterpretation caused by broader tuning curves in SNpr compared to other regions. The authors should address this more carefully in the manuscript.

We agree with the reviewer that this is an important point to investigate. In response to the reviewer's comment, we employed two approaches. First, we confirmed that our results are not solely dependent on our quantification of eye movement direction information. Second, we investigated the impact of tuning curve width on ω_p^2 using simulations, and tested whether it could account for our results.

To confirm that our results were not specific to the characteristics of ω_p^2 , we replicated our analysis using a classifier (similar to Fig. S4A-C; see Methods). The goal of the classifier was to determine the exact direction condition. The accuracy of the classifier was strongly correlated with the direction effect size (Spearman correlation: $p < 0.001$ for all populations and tasks), and exhibited the same trends for the saccade task (Fig. RR1A-C; Permutation Welch's ANOVA test: $p < 0.001$, Permutation Welch's t-test: $p_{\text{SNpr-caudate}}, p_{\text{SNpr-Purkinje}}, p_{\text{SNpr-local}} < 0.001$) and the pursuit task (Fig. RR1D-F; Permutation Welch's ANOVA test: $p < 0.001$, Permutation Welch's t-test: $p_{\text{SNpr-caudate}}, p_{\text{SNpr-Purkinje}}, p_{\text{SNpr-local}} < 0.001$). This was the case despite the fact that the variance of the SNpr was increased by the encoding of the reward probability condition (Fig. 4A). Classification errors may arise from the symmetry often found in tuning curves around the preferred direction of a neuron. If a neuron has a similar firing rate in two directions, a classifier will not be able to distinguish between them. To account for these kinds of errors, we calculated the average difference in distances between the predicted and actual directions from the preferred direction. For example, a neuron with a preferred direction at 0° , 45° and 315° would have the same distance from the preferred direction and therefore would be treated as the same class. The SNpr had the smallest differences, despite the fact that some SNpr neurons did not have a symmetric tuning curve or a well-defined preferred direction (not shown, saccade - Permutation Welch's ANOVA test: $p < 0.001$, Permutation Welch's t-test: $p_{\text{SNpr-caudate}}, p_{\text{SNpr-Purkinje}}, p_{\text{SNpr-local}} < 0.001$; pursuit - Permutation Welch's ANOVA test: $p < 0.001$, Permutation Welch's t-test: $p_{\text{SNpr-caudate}}, p_{\text{SNpr-Purkinje}}, p_{\text{SNpr-local}} < 0.001$)⁵.

Fig. RR1: Correlations of ω_p^2 effect sizes with the accuracy of a classifier in the motion epoch. Each dot represents a single neuron. In all plots, the horizontal position of the dots represents the ω_p^2 direction effect size. Bar plots show the averages and standard deviations of the mean for the y-axis. The r values represent the Spearman correlations. **A-C** show results from the saccade test and **E-F** show results from the pursuit task. **A** and **D** show neurons from the vermis. **D** and **E** show neurons from the basal ganglia.

We employed simulations to investigate the effect of tuning curve width on ω_p^2 . We constructed populations with Gaussian tuning curves, where we systematically manipulated their width (examples in Fig. RR2A). We kept the amplitude of the response constant (max-min difference), so that we could isolate the effect of width. We added Gaussian noise with varying standard deviations to mimic trial-by-trial variance. We repeated these simulations 1,000 times and calculated the average direction effect size for each width and added noise. We found that the tuning curve width had a slight, non-monotonous effect on the direction effect size (Fig. RR2B). To assess whether tuning curve width could account for our results, we identified combinations of different widths and added noise standard deviations that yielded comparable average effect sizes to those observed in our data. Fig. RR3C and D show that the observed effect sizes within the population could be generated by varying the underlying tuning curve widths. The colored dots in each figure represent areas in the matrix with similar effect sizes to the average direction effect size observed in the populations. The populations could not be separated based on the tuning curve width axis. This suggests that a similar width might potentially underlie the effect size across all populations. In contrast, the populations could be differentiated on the added noise standard deviation axis, indicating that this property plays a larger role in determining the effect size in comparison to the tuning curve width. We also simulated populations with Poisson noise and reached the same conclusions.

Fig. RR2: Simulation of Gaussian tuning curve populations with different widths. **A**, Examples of the population tuning curves used in the simulations. The different colors correspond to different tuning curve widths. **B**, Average direction effect size as a function of tuning curve width. The different colors correspond to different standard deviations of the added noise. **C** and **D**, Average direction effect size as a function of tuning curve width (horizontal) and noise standard deviation. Colored dots represent areas in the graph with a similar effect size as the average effect size observed in the different populations in the entire motion epoch. **C** shows data from the saccade task and **D** from the pursuit task.

1. Kostadinov, D. & Hausser, M. Reward signals in the cerebellum: origins, targets, and functional implications. *Neuron* (2022).
2. Bostan, A. C. & Strick, P. L. The basal ganglia and the cerebellum: nodes in an integrated network. *Nature Reviews Neuroscience* 2018 19:6 19, 338–350 (2018).
3. Caligiore, D. *et al.* Consensus Paper: Towards a Systems-Level View of Cerebellar Function: the Interplay Between Cerebellum, Basal Ganglia, and Cortex. *The Cerebellum* 2016 16:1 16, 203–229 (2016).
4. Nicholas, J. *et al.* The role of the cerebellum in learning to predict reward: evidence from cerebellar ataxia. *bioRxiv* 2022.11.04.515251 (2023) doi:10.1101/2022.11.04.515251.
5. Zur, G., Larry, N. & Joshua, M. High-Dimensional Encoding of Movement by Single Neurons in Basal Ganglia Output. *bioRxiv* 2023.05.17.541090 (2023) doi:10.1101/2023.05.17.541090.

REVIEWERS' COMMENTS

Reviewer #1 (Remarks to the Author):

The authors properly responded to all my comments. I recommend the publication of this study in Nature Communications.

Reviewer #2 (Remarks to the Author):

The authors have addressed all my concerns satisfactorily. I recommend publication.

Reviewer #3 (Remarks to the Author):

I appreciate the thorough responses to my concerns. I am satisfied with the new revision of the manuscript and have no further critiques.

Reviewer #1:

The authors properly responded to all my comments. I recommend the publication of this study in Nature Communications.

Reviewer #2:

The authors have addressed all my concerns satisfactorily. I recommend publication.

Reviewer #3:

I appreciate the thorough responses to my concerns. I am satisfied with the new revision of the manuscript and have no further critiques.

We thank the reviewers for their input throughout the revision process.